# Epidemiology and associated microbiota changes in deployed military personnel at high risk of traveler's diarrhea

William A. Walters[1], Faviola Reyes[2], Giselle M. Soto[3], Nathanael D. Reynolds[4], Jamie A. Fraser[5,6], Ricardo Aviles[2], David R. Tribble[5], Adam P. Irvin[7], Nancy Kelley-Loughnane[7], Ramiro L. Gutierrez[4], Mark S. Riddle[5], Ruth E. Ley[1], Michael S. Goodson[7]*, Mark P. Simons[4]

1 Max Planck Institute for Developmental Biology, Tuebingen, Germany, 2 Asociacion Benefica PRISMA, San Miguel, Peru, 3 U.S. Naval Medical Research Unit No. 6 (NAMRU-6),Callao, Lima, Peru, 4 Infectious Diseases Directorate, U.S. Naval Medical Research Center, Silver Spring, MD, United States of America, 5 Infectious Disease Clinical Research Program, Department of Preventive Medicine and Biostatistics, Uniformed Services University of the Health Sciences, Bethesda, MD, United States of America, 6 The Henry M. Jackson Foundation for the Advancement of Military Medicine, Inc., Bethesda, MD, United States of America, 7 711th Human Performance Wing, Air Force Research Laboratory, Wright-Patterson AFB, OH, United States of America

* michael.goodson.4@us.af.mil

**Data Availability Statement:** Sequence data are available in ENA under project PRJEB31759.

**Funding:** This work was funded by the Armed Forces Health Surveillance Branch Global Emerging

## Abstract

Travelers' diarrhea (TD) is the most prevalent illness encountered by deployed military personnel and has a major impact on military operations, from reduced job performance to lost duty days. Frequently, the etiology of TD is unknown and, with underreporting of cases, it is difficult to accurately assess its impact. An increasing number of ailments include an altered or aberrant gut microbiome. To better understand the relationships between long-term deployments and TD, we studied military personnel during two nine-month deployment cycles in 2015–2016 to Honduras. To collect data on the prevalence of diarrhea and impact on duty, a total of 1173 personnel completed questionnaires at the end of their deployment. 56.7% reported reduced performance and 21.1% reported lost duty days. We conducted a passive surveillance study of all cases of diarrhea reporting to the medical unit with 152 total cases and a similar pattern of etiology. Enteroaggregative E. coli (EAEC, 52/152), enterotoxigenic E. coli (ETEC, 50/152), and enteropathogenic E. coli (EPEC, 35/152) were the most prevalent pathogens detected. An active longitudinal surveillance of 67 subjects also identified diarrheagenic *E. coli* as the primary etiology (7/16 EPEC, 7/16 EAEC, and 6/16 ETEC). Eleven subjects were recruited into a nested longitudinal substudy to examine gut microbiome changes associated with deployment. A 16S rRNA amplicon survey of fecal samples showed differentially abundant baseline taxa for subjects who contracted TD versus those who did not, as well as detection of taxa positively associated with self-reported gastrointestinal distress. Disrupted microbiota was also qualitatively observable for weeks preceding and following the incidents of TD. These findings illustrate the complex etiology of diarrhea amongst military personnel in deployed settings and its impacts on job performance. Potential factors of resistance or susceptibility can provide a foundation for future clinical trials to evaluate prevention and treatment strategies.

Infections Surveillance (GEIS) program (MPS), the Military Infectious Diseases Research Program (MIDRP) (MPS), the 711th Human Performance Wing Research, Studies, Analysis and Assessment Committee [NKL, MSG], and the Max Planck Society [WAW, REL]. One of the coauthors (JAF) is affiliated with the non-profit Henry M. Jackson Foundation. The Henry M. Jackson Foundation only provided JAF with a salary, and did not have any role in our study design, execution, or manuscript preparation. The funders had no role in study design, data collection and analysis, decision to publish, or preparation of the manuscript.

**Competing interests:** One of the coauthors, Jamie Fraser, is affiliated with the non-profit Henry M. Jackson Foundation. The Henry M. Jackson Foundation only provided Fraser with a salary, and did not have any role in our study design, execution, or manuscript preparation.

## Introduction

Travelers' diarrhea (TD) remains a major risk to deployed military forces worldwide, in addition to its impact on civilian travelers and mobile populations [1]. As an example of the potential impact on military operations, a study to evaluate health outcomes for U.S. forces deployed to Egypt in 2005 for Operation Bright Star (a biannual exercise with the Egyptian armed forces) revealed that 40% of troops experienced TD during this operation, but only 4% sought formal medical care [2]. An anonymous questionnaire of over 15,000 U.S. personnel deployed to Iraq or Afghanistan in 2003–2004 found that 78.6% of troops in Iraq and 54.4% in Afghanistan experienced GI-related illnesses, with 80% seeking care from their unit medic [3]. Eating local food from non-U.S. sources was associated with an increased risk of illness [4]. Additionally, over 50% of travelers were affected by TD during a two week visit to a developing country [1]. Due to their mission-aborting potential, the U.S. military has placed a high priority on the development of effective methods to prevent the most common enteropathogenic diseases [5]. Bacterial pathogens, such as enterotoxigenic *Escherichia coli* (ETEC), *Campylobacter jejuni*, and *Shigella* spp. (particularly *S. flexneri* and *S. sonnei*) are often the most frequent causes of TD in both adults and children, with variability among regions around the world [1,5,6]. However, while common diarrhea-causing pathogens have been implicated in the majority of TD cases, a large proportion of cases have no known cause [6,7]. The role of the gut microbiome, both in terms of harboring potential TD-causing pathogens and providing either protection or susceptibility to TD, has not been fully explored.

The normal constituents of the human gut microbiome are primarily bacteria, but also include archaea, eukaryotes, and viruses. These microbial cells reach levels of billions per milliliter in the large intestine [8]. Each person carries their own personalized mix of different microbes that have been acquired over a lifetime and whose diversity was built up from birth via exposure to family members, food, and other environmental sources [9]. Collectively, these microbes are equal to, or outnumber the cells in the human body [10], and encode 300 times more genes than the human genome [11]. These microbes interact closely with the host immune system [12], and they perform critical services for the human host, such as enhanced degradation of dietary components, production of vitamins, degradation of xenobiotics, and protection from pathogen invasion [13]. An increasing number of diseases stem from an altered and aberrant microbiome, ranging from inflammatory bowel disease to diabetes [14]. There is evidence that the gut microbiome plays a role in TD [15,16], and that TD can change the gut community structure[17–19]. Thus, we hypothesize that gut microbial community composition can be a significant factor in symptomatic and sub-clinical syndromes, and can contribute to susceptibility or resilience to TD.

Gaining a better understanding of how microbial community composition affects military personnel experiencing diarrhea during deployment is important for strategic decisions regarding potential medical interventions, such as use of probiotics as prophylaxis and/or post-infectious therapy. In this study, we initiated a longitudinal cohort surveillance study of 67 individuals that were followed during two 9-month deployment cycles in 2015–2016 to Honduras. After enrollment and collection of baseline samples, subjects were contacted weekly to assess presence of any ailment, including diarrhea and associated symptoms, and if ill, were asked to complete a case report form and submit a stool sample for testing. Additionally, 11 subjects were recruited into a nested substudy to specifically examine gut microbiome composition and changes associated with deployment and illness, with these subjects voluntarily submitting stool samples weekly, regardless of illness, and completing a daily diet, activity, and well-being log.

## Materials and methods

### Study site and fieldwork

We developed a surveillance site at a forward operating (frontline) military base in Honduras. The base had a population of approximately 600 active-duty and reserve military personnel assigned on one year, six months or 90 day orders. Military personnel were deployed to the site at various times throughout the year on a rolling basis. These individuals frequently arrived as units and remained onsite for the duration of the orders assigned to the unit, and until their replacements arrived. All personnel were prescribed doxycycline for daily use as an antimalarial prophylactic. An on-site medical unit staffed by active duty and reserve personnel and several local Honduras physicians provide year-round medical care to military personnel. While there was a fixed dining facility on base that was monitored frequently for sanitation and food safety, personnel had access to the local economy during their official duties outside the base and when off-duty they were able to visit local towns. TD was defined as three or more loose stools in a 24 hr period. Rates of TD at this location throughout deployment, especially during the first weeks to first month on post, were comparable to rates experienced by military personnel in El Salvador (27%) [20] and Peru (24.8%) [21].

We implemented two studies to investigate TD during deployment: Study 1 was a passive surveillance study encompassing subjects that reported to the clinic with TD and end-of-deployment questionnaires (N = 1172); and Study 2a was an active surveillance study involving subjects that were recruited for active follow up after they arrived on post (N = 67). The longitudinal microbiome study, where subjects voluntarily supplied weekly fecal samples and entered health and diet information into a daily log, was nested within Study 2a and is termed Study 2b (N = 11). All studies were conducted from February 2014 to November 2016.

**Study 1.** For individuals to be granted Sick in Quarters (SIQ) status when ill they must see a medical provider and provide documentation of their illness to their unit leadership, therefore those reporting to the clinic seeking care often had more severe symptoms. A stool sample was collected for onsite laboratory testing. To capture mild to moderate cases that are typically not reported or captured, we also integrated questionnaires into medical out-processing for all base personnel at the end of each unit's deployment cycle, similar to a prior mid-deployment TD questionnaire-based study [22]. The questionnaire is available as S1 File. The questionnaires asked about past events including signs and symptoms of diarrhea, respiratory, and febrile illnesses during their deployment, and included questions regarding the impacts of illness on job performance and healthcare seeking outcomes.

**Study 2a.** A briefing to explain the study details was performed by non-military study personnel as part of the routine medical in-processing. After a short description of the study, a copy of the consent form was left for individuals to review and if they decided to participate they contacted the study staff individually. There were no exclusion criteria, apart from a requirement of a minimum time on base of one month. Subjects were individually consented to participate in the study and afterwards completed a cohort enrollment form containing questions to collect general demographic data. Potential signs and symptoms of diarrhea and respiratory disease were also captured. The onsite study personnel contacted the subjects once a week by email or by phone to inquire if they have had any diarrhea illness during the last week. If the subject's symptoms met TD definitions, the subject was requested to come to the medical unit and complete a brief case report form regarding their current symptoms. The subject was given a stool collection kit and a stool sample was provided at the subject's convenience and returned to the laboratory for immediate processing and onsite laboratory testing.

**Study 2b.** Within the longitudinal cohort design of Study 2a, we also nested a cohort substudy of 11 individuals to specifically examine gut microbiome changes associated with

deployment and illness. In addition to the other components of the active cohort study, these 11 subjects voluntarily submitted stool samples weekly regardless of illness, case report forms, stools for every diarrhea episode, and completed a daily diet, activity, and well-being log on an individual Apple iPod, as described below.

This study was jointly approved by the Institutional Review Board at U.S. Naval Medical Research Unit No. 6 (NAMRU-6) in Lima, Peru and 711th Human Performance Wing, Air Force Research Laboratory at Wright Patterson Air Force Base in Ohio. Additionally, the study was approved by both the military base Commander and the regional Medical Officer.

## Samples and laboratory analysis

**1) Study 1 and 2a.** Demographic Data: Subjects were analyzed on the basis of demographic data (rank, sex, branch of service, occupation, deployment duration, travel, vaccinations, preventive medicine guidance, other gastrointestinal disorders, drugs and antibiotics taken, and symptoms within the prior two weeks and at enrollment). Continuous data (e.g. days of decreased job performance) were compared using analysis of variance, followed by individual comparisons using the Bonferroni correction. Differences in distributions of categorical data were compared using the Chi-square test or Fisher's exact test, if Chi-square assumptions were not met.

Incidences of TD were calculated using total time during study participation and the duration of each TD episode. These data, in combination with all laboratory data, were used to calculate incidences for each pathogen. STATA v13 (StataCorp LLC, College Station, TX) was used for analysis of the survey and demographic data.

*Stool analysis.* After self-collection of stools (commode specimen collection system, Fisher Scientific, St. Louis, MO) by subjects, the samples were transported to the laboratory located in the same building as the medical unit and were immediately cultured for bacterial enteropathogens by conventional microbiological techniques [23,24]. Stool specimens were streaked directly onto xylose lysine deoxycholate (XLD), thiosulfate citrate bile salt sucrose agar (TCBS), MacConkey agar, Salmonella-Shigella agar, and Campylobacter blood agar [25]. Inoculated Campylobacter blood agar plates were incubated at 42°C for 48 h in a microaerophilic atmosphere. Bacterial cultures were examined for the following agents at 24, 48, and 72 h for *Salmonella* and *Shigella* spp. Identifications were performed using API-20E identification strips (Biomerieux, Durham, NC, USA). Aliquots of stool samples and all identified stool culture isolates were frozen at -80°C and shipped commercially to U.S. Naval Medical Research Unit No. 6 (NAMRU-6) in Lima, Peru for confirmatory testing. Upon arrival at the laboratory, samples were stored at -80°C until tested. Stool aliquots were fixed in sodium acetate-acetic acid-formalin (SAF) which included 10% formalin (prepared in-house) and were sent to NAMRU-6. These samples were processed for examination for protozoal pathogens (*Entamoeba histolytica, Giardia lamblia,* etc) using modified-Trichrome stain of fecal smears, as well as examined for helminth ova by wet mount following concentration of the samples [26]. In addition, fecal smears were stained using the modified Kinyoun acid-fast method for the identification of *Cyclospora, Cryptosporidium,* and *Isospora.*

## Molecular detection of enteropathogens

In addition to bacterial culture, all stools from ill subjects were tested using the Biofire FilmArray GI panel (Biofire, Salt Lake City, UT, USA) for detection of the pathogens enteroaggregative *E. coli* (EAEC), enteropathogenic *E. coli* (EPEC), enterotoxigenic *E. coli* (ETEC) lt/st, Shiga-like toxin-producing *E. coli* (STEC) stx1/stx2, *E. coli* O157, *Shigella*/Enteroinvasive *E. coli* (EIEC), *Campylobacter* (*jejuni, coli,* and *upsaliensis*), *Clostridium difficile* (toxin A/B),

*Plesiomonas shigelloides*, *Salmonella*, *Yersinia enterocolitica*, *Vibrio* (*parahaemolyticus*, *vulnificus*, and *cholerae*), viruses [Adenovirus F40/41, Astrovirus, Norovirus GI/GII, Rotavirus A, Sapovirus (I, II, IV, and V)], and parasites [*Cryptosporidium*, *Cyclospora cayetanensis*, *Entamoeba histolytica*, *Giardia lamblia*]. Stools were processed and tested on the FilmArray GI panel according to the manufacturer's procedures. Intersecting sets of pathogens were generated via the UpSet analysis tool [27].

## Study 2b: Microbiome collection, processing, and analyses

The subjects (N = 11) provided weekly fecal samples, when possible, on site at the medical unit. Stools were self-collected by subjects utilizing the commode specimen collection system (Fisher Scientific, St. Louis, MO). These were transferred to onsite study personnel where samples were separated into 5 mL aliquots in cryovials and stored at -80˚C. Samples were intermittently shipped to Tuebingen, Germany via Wright-Patterson Air Force Base on dry ice. These samples were stored at -80˚C until processing. DNA was extracted from these samples using the MagAttract PowerSoil DNA kit (QIAGEN, Hilden, Germany) with the kit's protocol for automated liquid handling systems. Next a two stage PCR process was done to amplify and barcode the hypervariable V4 region of the 16S ribosomal RNA small subunit gene. Briefly, the 515f and 806r primers [28] with Nextera Transposase Adapters (Illumina Corp., San Diego, California, USA) were used for PCR amplification at 25 cycles. Then these amplicons then went through an 8 cycle PCR with the Nextera Index primers (using 6 base pair dual barcodes), followed by clean-up using the Mag-Bind PCR clean-up 96 well kit (Omega Biotek). The resulting libraries were sequenced using 250 bp paired-end reads on an Illumina MiSeq system at the Genome Center for the Max Planck Institute for Developmental Biology in Tuebingen, Germany. Negative controls (14 PCR water blanks) and positive control (9 stool samples from the same donor, an individual from the USA) were included and filtered from the final results, as the negative controls had low sequence counts, and did not cluster near adjacent wells in PCoA plots, while the positive controls showed expected behavior (consistent taxonomies and within-subject clustering).

E-survey: To gather self-reported data, the subjects were provided with iPod touch systems (Apple) that had a questionnaire covering diet, daily activities, and self-reported ailments, including self-reported 'gastrointestinal issues'. A spreadsheet of the questionnaire data (manual corrections were done to have consistent date and time format) are available as S1 Table. Since a variable number of questionnaire entries could be matched to a weekly stool sample, a custom Python script (described below) was used to match questionnaire data to stool data (i.e., entries from the same day or up to six days prior) and average the data, if any, across that time period to match the stool data.

Sequencing data were processed using the QIIME 2 (2018.8 release) [29] and R [30] software packages (the R environment is listed in the S1 Appendix). A QIIME metadata mapping file is available as S2 Table. DADA2 [31] was used to process the paired-end reads, and generate sequence variants (SVs) with the following parameters:—p-trim-left-f 19,—p-trim-left-r 20,—p-trunc-len-f 210, and—p-trunc-len-r 210. The SILVA [32] 132 release (99% OTUs) was used for taxonomic assignments. Next, taxonomy was filtered out that matched these strings: 'D_0__Eukaryota,D_4__Mitochondria,D_3__Chloroplast'. The filtered SVs were aligned using MAFFT [33] with default parameters, the resulting alignment was filtered using the—p-max-gap-frequency 0.80 parameter, and a phylogenetic tree was inferred using FastTree with midpoint rooting. Samples with less than 4617 sequences were filtered from the data. Beta diversity and alpha diversity were calculated with the default metrics/measures: weighted and unweighted UniFrac, Bray-Curtis, and Jaccard for beta diversity and Chao1, Faith's PD, evenness, shannon, and observed OTUs for alpha diversity [34–38]. With specific time points of

TD removed, volatility, i.e., fluctuations away from starting values (either alpha diversity values or distances over time for beta diversity) were tested using the QIIME 2 longitudinal plug-in. The "first-differences" function was used in the case of alpha diversity, and "first-distances" was used for beta diversity metrics/measures. Order (see metadata mapping file) was used for the—p-state-column parameter and—p-individual-id-column was Subject. Significance was tested using the "linear-mixed-effects" function of the longitudinal plug-in, with "—p-group-columns SubjectHadTD" as the tested groups.

Linear mixed models for differential abundance were performed using the lmer4 package in R [30,39]. Specific time points of TD were filtered from the data, and zero-inflation was minimized by filtering out SVs that had less than 1000 sequences or were found in less than 10 samples, leaving 389 SVs. Additionally, subject 34 was removed, as there were only three samples from this subject, and two were filtered (one for low SV count, and the other was a TD time point). The test and null model were:

model <- lmer(log10(Abundance^(1/3)+1)~SubjectHadTD + standardized_counts + (1|Subject) + (1|PlateNumber) + Order, data = curr_data, REML = FALSE)

null <- lmer(log10(Abundance^(1/3)+1)~standardized_counts + (1|Subject) + (1|PlateNumber) + Order, data = curr_data, REML = FALSE)

where Abundance is the counts of reads, SubjectHadTD is TRUE/FALSE for the subject according to whether any TD incidents occurred, standardized_counts are the sequence counts standardized via the decostand function (using "standardize") of the R vegan package [40], the Subject is the participant, PlateNumber is the plate that the samples were processed on, and Order is the sample order.

The test and null model were compared with the R anova test, to determine if the test model fit the data significantly better than the null model. Only the SVs with residual histograms and Q-Q plots that reflected normality were reported as significant in S3 Table. Differential alpha diversity values were tested in a similar manner (for Chao1, ObsOTUs, PD, and Shannon metrics) for TD+ versus TD- subjects, with the average alpha diversity value for 10x rarefactions at 4617 sequences/sample used instead of SV Abundance values (no data transformation necessary for normality). To compare self-reported iPod questionnaire data to microbiome data, first the (up to daily) entries had to be related to a collected fecal sample (up to weekly). To do this, data were converted into quantitative form (e.g., a "true" entry for GI distress becomes 1, "false" becomes 0) and the data were averaged for the day of the fecal sample and the six prior days. The custom Python script is available on gisthub (https://gist.github.com/walterst/ca4a41d32cceba809c77b55fc2c068cc). The QIIME-formatted metadata mapping file, S2 Table, includes the parsed data from the iPod questionnaire. Because not all samples have associated iPod questionnaire data (137 of 215 samples had iPod data in range, see MatchingIPodData field in S2 Table), samples lacking data had to be filtered before further testing. A linear mixed model was applied in a similar fashion to the categorical tests for SVs, above, for self-reported Gastrointestinal issues across all SVs:

model <- lmer(log(Abundance^(1/6)+1)~Gastrointestinal_issues + standardized_counts + (1|Subject) + (1|PlateNumber) + Order, data = curr_data, REML = FALSE)

df.null <- lmer(log(Abundance^(1/6)+1)~standardized_counts + (1|Subject) + (1|PlateNumber) + Order, data = curr_data, REML = FALSE)

where Gastrointestinal_issues is the self-reported GI distress values from the iPod questionnaire, while other variables are as described in the above section for differential abundance. A total of five SVs had p-values < 0.05 after FDR (False Discovery Rate, Benjamini-Hochberg), but only the two SVs had residual histograms and Q-Q plots that reflected normality are reported on S3 Table. Significance after FDR correction, for stool consistency and time slept, was not detected.

To compare the clustering of the subjects in this study with geographically distinct subjects, the V4 16S ribosomal RNA small subunit amplicon sequence data from a global gut survey [41] were randomly subsampled to 3% of the total reads, which approximates an Illumina MiSeq run at ~30000 reads/sample. The random subsampling Python script is available on gisthub (https://gist.github.com/walterst/22c9fb9d1f817eae55c14a84b1b106d9). As the sequences were not the same lengths and were generated with different sequencing technologies (making a sequence variants approach inappropriate), the read data for both the Yatsunenko study and the reads for the TD subjects were clustered in a "closed-reference" approach, at 97% identity, against the SILVA 132 97% OTUs. Beta diversity was calculated with the weighted UniFrac metric at an even sampling of 4257 sequences per sample. Samples were filtered so that only adult subjects were retained. The combined mapping file for the TD subjects and the subjects from the Yatsunenko study is available as S4 Table.

The Firmicutes:Bacteroidetes ratio was calculated by first averaging each individual subject's Firmicutes and Bacteroidetes relative abundances (with TD time points removed) as a per-subject average, then averaging the TD+ versus TD- subject averages.

Sequence data are available in ENA under project PRJEB31759.

## Results

### Epidemiology studies

A total of 1,172 individuals completed the questionnaires from 2014–2016 as part of study 1. The majority of the personnel completing the questionnaires were male (82.6%) with a mean age of 32.7 (32.3, 33.3, 95% CI) years for all subjects. The mean age of female subjects was slightly older (33.5 years) than males (32.7 years, Table 1). Reporting of military rank showed the majority of subjects were enlisted personnel (71.8%). The majority of the personnel deployed to this site were reserve forces, whereas the command leadership was primarily active duty. In review of the occupations most of the subjects were medical personnel (not shown). Of those completing the questionnaires, 293/1173 (25.0%) reported experiencing TD during their deployment, with a median of two episodes per person reported with each episode lasting a median of two days (Table 2). Additionally, 48.3% reported experiencing moderately severe TD defined at 3–5 stools per 24h period, 42.9% reported severe TD defined as greater than or equal to six stools per 24h period, and only 8.4% with mild TD defined as 1–2 stools per 24h period. For duty days lost, 78.8% subjects who completed the questionnaires reported no time lost (Table 2). However, 57.6% reported experiencing a reduction in their ability to perform their job from their illness, ranging from 1–60 days. For questions that asked about care-seeking behaviors associated with illness, 38.7% reported to urgent care (sick call) at the clinic and 15.2% reported being in a Sick in Quarters (SIQ) status and unable to work (Table 2). Similarly, 22.1% reported taking antibiotics as a treatment for illness, although we were not able to

**Table 1. Post deployment questionnaire demographics.**

| Subjects | n | Mean age (95% CI) |
|---|---|---|
| Female | 203 (17.4%) | 33.5 (32.3, 34.7) |
| Male | 965 (82.6%) | 32.7 (32.2, 33.2) |
| Enlisted | 842 (71.8%) | |
| Officer | 329 (28.1%) | |
| Total | 1172* | 32.8 (32.3, 33.3) |

*33 individuals did not report Gender

**Table 2. Post deployment questionnaire, impact of diarrhea.**

| Variable (number of respondents) | Result | n (%) |
|---|---|---|
| Diarrhea (1172) | Yes | 293 (25.0%) |
| | No | 879 (75.0%) |
| | Median episode # per person (min, max) | 2 (1, 40) |
| Stools/24 h (443) | Mild (1–2 stools) | 37 (8.4%) |
| | Moderate (3–5 stools) | 214 (48.3%) |
| | Severe (>6 stools) | 190 (42.9%) |
| Days experiencing diarrhea (445) | Median per person (min, max) | 3 (1, 60) |
| Duty days lost (448) | 0 days | 353 (78.8%) |
| | 1–2 days | 81 (18.1%) |
| | 3–40 days | 14 (3.1%) |
| Days reduced performance (448) | 0 days | 190 (42.4%) |
| | 1–5 days | 234 (52.2%) |
| | >5 days | 24 (5.4%) |
| Clinical follow up (447) | Sick call | 173 (38.7%) |
| | Sick in quarters | 68 (15.2%) |
| | Intravenous fluid | 42 (9.4%) |
| | Antibiotic | 99 (22.1%) |

determine whether the prescription was given at the clinic or prior to deployment from their travel medicine clinicians and self-administered. It should be noted that all subjects were mandated to take doxycycline as an antimalarial prophylactic during deployment. Only 9.4% reported needing intravenous (IV) fluids during their clinical care. We collected data on symptoms associated with TD during deployment, including those experiencing nausea (35.8%), vomiting (14.8%), fever (28.2%), bloody diarrhea (4.9%), headache (33.3%), abdominal cramps (58.1%), and joint or muscle aches (28.2%) (Table 3).

Through the period of February 2014 through November 2016, 152 subjects reported to the clinic with complaints of diarrhea and were enrolled in the passive surveillance study. These subjects completed a case report and provided a stool for laboratory workup. Stool culture detected 2 cases of *Shigella* and 1 case of *Salmonella* infections while an *ad hoc* ova and parasite exam did not find any cases of protozoan or helminth infections among those enrolled (Table 4). However, implementation of Biofire FilmArray GI assays detected pathogens in 90.8% of cases (Table 4), of which 41.4% had multiple pathogens detected. This culture-independent detection provided information on TD that had not been captured among deployed military personnel previously. Our findings showed that diarrheagenic *E. coli* pathogens were

**Table 3. Symptoms associated with diarrhea.**

| n = 445 | Yes | No |
|---|---|---|
| Nausea | 160 (35.8%) | 287 (64.2%) |
| Vomiting | 66 (14.8%) | 379 (85.2%) |
| Fever | 126 (28.2%) | 321 (71.8%) |
| Bloody diarrhea | 22 (4.9%) | 136 (95.1%) |
| Headache | 149 (33.3%) | 298 (66.7%) |
| Abdominal cramps | 259 (58.1%) | 187 (41.9%) |
| Joint/muscle aches | 126 (28.2%) | 321 (72.8%) |

*Not all questions received responses and had missing data

**Table 4. Study 1 Laboratory results. O & P indicates ova and parasites.**

| Method | Result | n | % |
|---|---|---|---|
| Culture | *Shigella* | 2 | 1.3 |
| | *Salmonella* | 1 | 0.7 |
| O & P | Negative | 152 | 100 |
| FilmArray | EAEC | 52 | 34.2 |
| | ETEC | 50 | 32.9 |
| | EPEC | 35 | 23.0 |
| | STEC | 20 | 13.2 |
| | *Shigella*/EIEC | 18 | 11.8 |
| | Norovirus | 12 | 7.9 |
| | *Cryptosporidium* | 7 | 4.6 |
| | *Campylobacter* | 5 | 3.3 |
| | *Cyclospora* | 3 | 2.0 |
| | *Clostridium difficile* | 3 | 2.0 |
| | *Yersinia enterocolitica* | 1 | 0.7 |
| | Rotavirus | 1 | 0.7 |
| | Multiple pathogen (2–6) | 63 | 41.4 |
| | Total Positive | 138 | 90.8 |
| | Total | 152 | |

EAEC: enteroaggregative E. coli, EPEC: enteropathogenic E. coli, ETEC: enterotoxigenic E. coli, STEC: Shiga-like toxin producing E. coli, EIEC: Enteroinvasive E. coli.

predominant at this site, with EAEC as the most common (34.2%), followed by ETEC (32.9%), EPEC (23.0%), and STEC (13.2%). *Shigella* was detected in 18 cases (11.8%) compared to only 2 cases (1.3%) found by bacterial culture, highlighting the increased sensitivity of the non-culture base FilmArray method. Other pathogens detected included norovirus (7.9%), *Cryptosporidium* (4.6%), *Campylobacter* species (3.3%), *Cyclospora* (2.0%), *Clostridium difficile* (2.0%), and one case each of *Yersinia enterocolitica* and rotavirus. These findings show the complexity of TD among deployed military personnel and highlight the challenges in prevention and treatment efforts for multi-microbial infections of bacterial, viral, and protozoan pathogens.

To better understand the risk and impact of TD at the military base, we also conducted an active surveillance study (study 2a) in tandem with the passive surveillance (study 1) and post-deployment questionnaires. The study was a longitudinal cohort design with subjects recruited at the start of their deployment. In this study, we were able to recruit 67 total subjects over two deployment cycles from 2015–2016. The majority of subjects were males (74.6%) with a mean age of 37.6 years (Table 5). Analysis of data collected from subjects at the time of enrollment showed no significant trends or associations. Of note, 100% of those enrolled lived on-base for the study, 57% noted traveled in the month prior, and 5/67 (7.7%) noted experiencing diarrhea and/or vomiting in the previous two weeks with 4/67 (6.0%) reporting diarrhea at the time of enrollment. Additionally, 55/65 (84.6%) reported taking an antibiotic, of which doxycycline or

**Table 5. Study 2a cohort demographics.**

| Subjects | n | Mean age (95% CI) |
|---|---|---|
| Female | 17 (25.4%) | 42.2 (36.9, 47.6) |
| Male | 50 (74.6%) | 37.6 (34.7, 40.5) |
| Total | 67 | |

**Table 6. Study 2a diarrheal incidence.**

| Variable | |
|---|---|
| Time at risk | 568.8 person months |
| Diarrhea episodes | 17* |
| Incidence rate | 2.99 per 100 person months |

*Only 16 samples received for laboratory testing

another antimalarial drug was listed. The total time the population was at risk was 568.8 person-months and during this time 17 episodes of diarrhea were reported to the study coordinator for an incidence rate of 2.99 cases per person-month (Table 6). For comparison, 11 cases of respiratory disease were detected for an incidence rate of 1.93 cases per person-month. Upon reporting of a case of diarrhea, subjects were asked questions regarding specific risk factors including drinking tap water and eating at alternative locations such as restaurants, hotels, and street vendors but due to the small sample sizes no significant associations with illness were found. Of the 17 diarrheal events, 16 stool samples were submitted for laboratory workup. *Shigella* was found in only one case, by bacterial culture, with no protozoa or helminths detected (Table 7). The FilmArray GI panel was positive for 12/16 (75.0%) of samples tested with 9/16 found to have multiple pathogens. Similar to the findings from our passive surveillance cases, diarrheagenic *E. coli* pathogens were most predominant with EAEC and EPEC as the most common (43.8% equally), followed by ETEC (37.5%), and STEC (18.8%). Other pathogens detected included rotavirus, *Cyclospora*, and *Campylobacter* (Table 7). Co-infections predominated (~63% of cases, Fig 1). While pathogens were detected as mono-infections, a majority of the time, each pathogen was detected as a co-infection with one or more of the diarrheagenic *E. coli* strains. This includes the *E. coli* strains themselves—for example, ETEC was detected alone in 15 cases, but it was detected alongside another *E. coli* strain in 40 cases.

## Microbiome changes were associated with deployment and TD events

The 11 subjects recruited into the microbiome cohort had an average age of 36.3 (23.7, 48.9; 95% CI), with deployments ranging from 147–466 days (Table 8). Four experienced diarrhea, with two of these subjects testing positive for enteric pathogens on the Biofire array assay.

**Table 7. Study 2 laboratory results. O & P indicates ova and parasites.**

| Method | Result | n | % |
|---|---|---|---|
| Culture | *Shigella* | 1 | 6.3 |
| O & P | Negative | 16 | 100 |
| FilmArray | EPEC | 7 | 43.8 |
| | EAEC | 7 | 43.8 |
| | ETEC | 6 | 37.5 |
| | STEC | 3 | 18.8 |
| | *Campylobacter* | 1 | 6.3 |
| | *Cyclospora* | 1 | 6.3 |
| | Rotavirus | 1 | 6.3 |
| | Multiple pathogen (2–6) | 9 | 56.3 |
| | Total Positive | 12 | 75.0 |
| | Total | 16 | |

EAEC: enteroaggregative *E. coli*, EPEC: enteropathogenic *E. coli*, ETEC: enterotoxigenic *E. coli*, STEC: Shiga-like toxin producing *E. coli*, EIEC: Enteroinvasive *E. coli*.

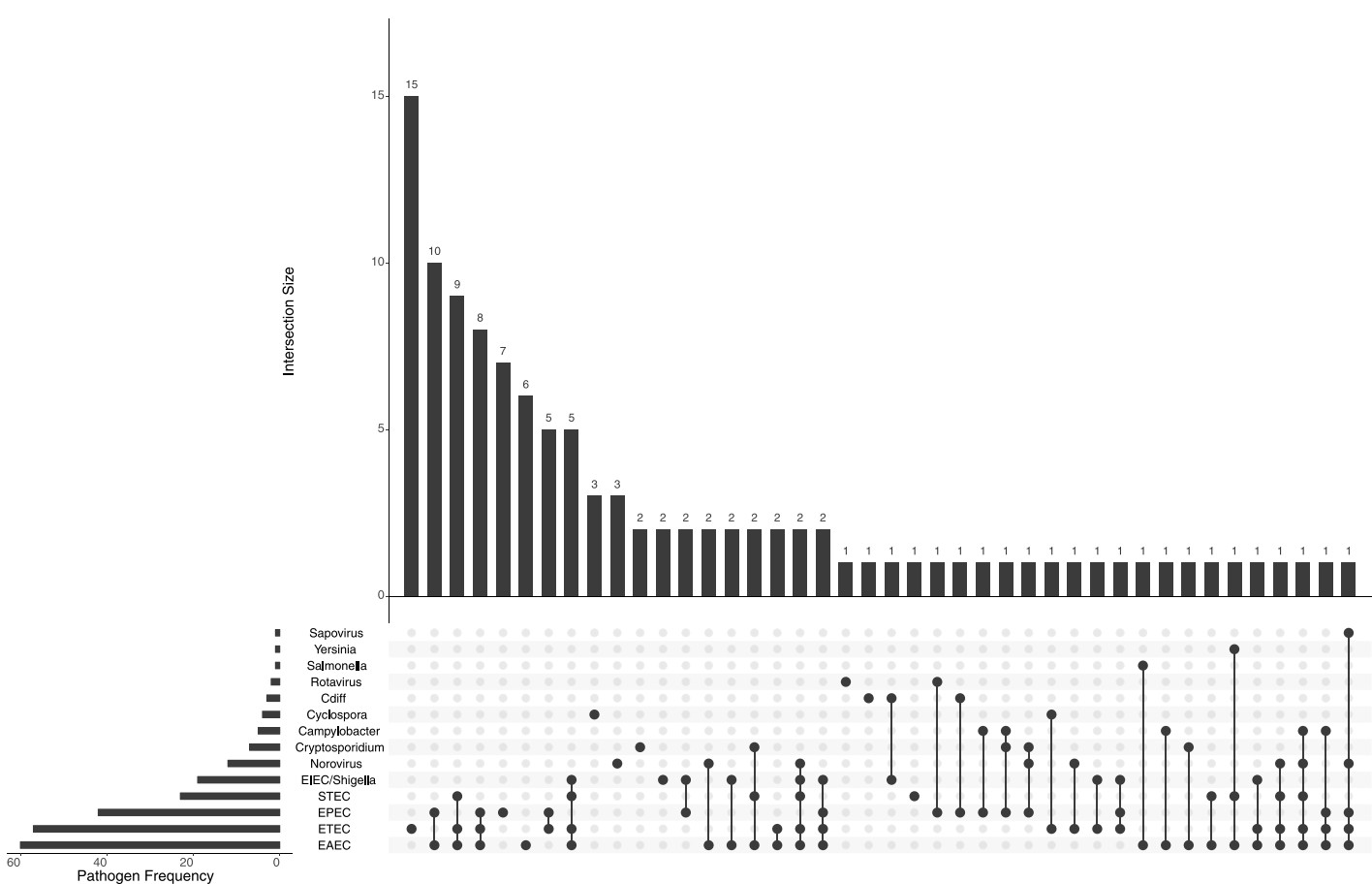

**Fig 1. Intersections of pathogen infections across the study 1 and 2 cohorts.** Pathogens are shown on the lower left, with the size of the horizontal bars indicating the frequency of the pathogen incidents across the data set. The vertical bar charts indicate the sorted counts of each set interaction, with the bottom dots and connecting lines indicating detection of a pathogen and its co-occurrence with other pathogens, e.g., ETEC occurred alone 15 times, EPEC and EAEC were a co-infection 10 times, and so on. Co-infections were quite prevalent, with rare cases of predominantly singly-infectious agents, such as Cyclospora (detected three times alone, once as a co-infection with ETEC).

**Table 8. Nested cohort (Study 2b), demographics and summary of diarrheal incidents and clinical results.**

| Subject | Age | Sex | time at risk (days) | time at risk (months) | episodes diarrhea | Pathogens |
|---|---|---|---|---|---|---|
| 020 | 20 | M | 147 | 4.9 | 0 | N/A |
| 022 | 21 | M | 148 | 4.93 | 0 | N/A |
| 023 | 32 | M | 144 | 4.8 | 0 | N/A |
| 024 | 34 | M | 157 | 5.23 | 0 | N/A |
| 025 | 30 | M | 158 | 5.27 | 0 | N/A |
| 029 | 47 | F | 149 | 4.97 | 4 | EPEC/ ETEC/ EAEC/Cyclospora |
| 033 | 33 | M | 146 | 4.87 | 2 | EAEC/ETEC/ STEC/ E.coli 0157 |
| 034 | 50 | M | 147 | 4.9 | 1 | Not detected |
| 035 | 56 | F | 154 | 5.13 | 0 | N/A |
| 041 | NA | NA | 466 | 15.53 | 0 | N/A |
| 043 | 40 | M | 365 | 12.17 | 1 | Not detected |

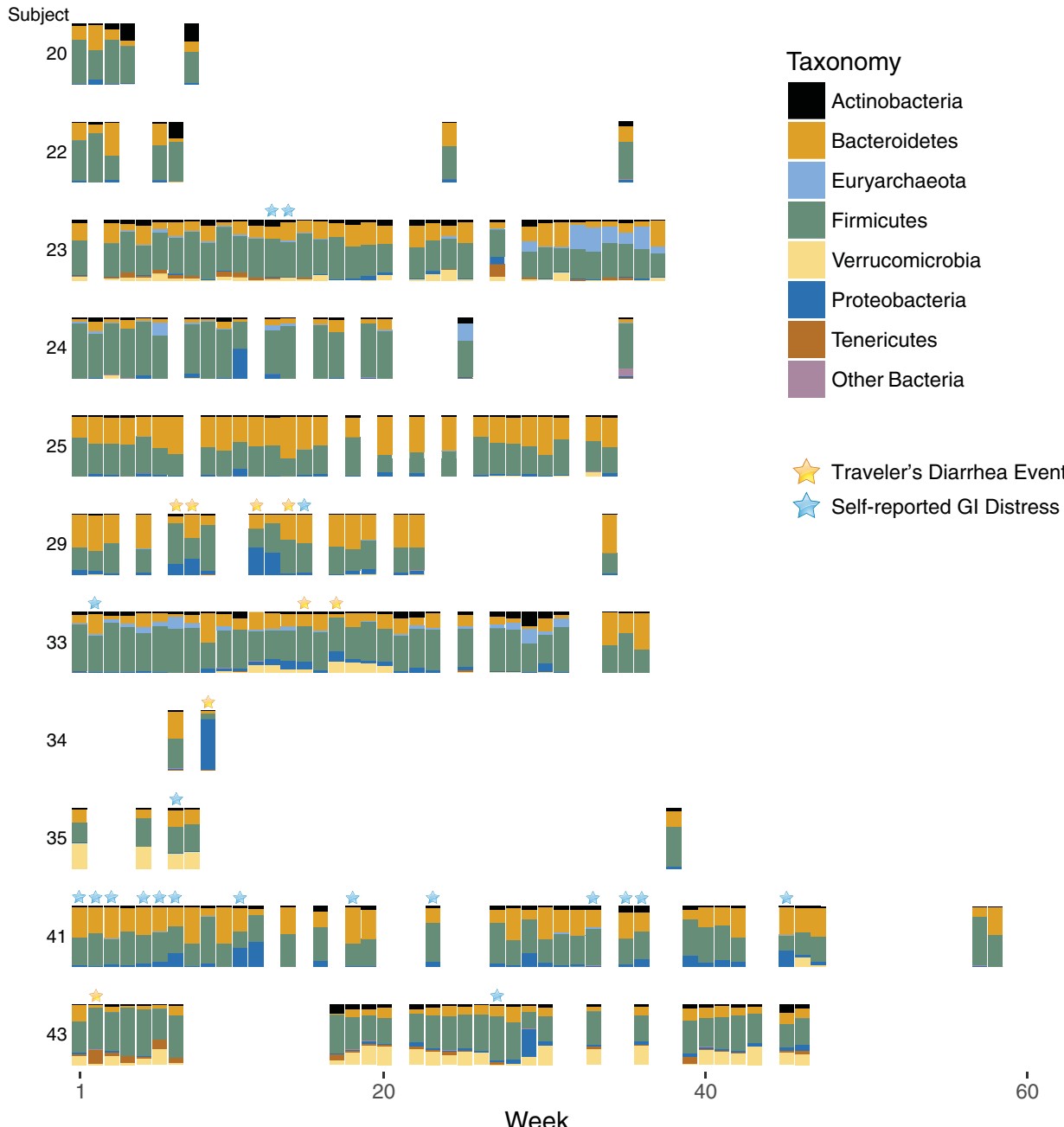

**Fig 2. Study 2b Microbial phyla over time for all subjects.** The relative abundance of the most abundant phyla are shown for each subject along with the weekly time points where samples were collected. Week 1 indicates the first week of deployment overseas. TD events are indicated with a gold star and self-reported GI distress is indicated by blue stars.

Fecal 16S ribosomal RNA small subunit hypervariable region V4 amplicons were grouped into identical sequence variants (SVs). The relative abundance of the dominant phyla, derived from these SVs, arranged according to sampling week and with incidents of TD or self-reported GI distress, are shown in Fig 2. There is sparsity in the week to week voluntary sampling, and variability in the number of samples provided by subject, however, there are trends and observations of interest. In general, subjects had their own microbial signature that

remained mostly consistent over time. Fluctuations in certain taxa do not appear to be related to TD or GI distress events, e.g., Actinobacteria increases substantially in relative abundance for subjects 20, 22, and 33, without any apparent relationship to TD, and subject 23 is dominated in late (week 30+) time points with Archaea (*Methanobrevibacter smithii*) with no reported clinical disease (self-reported data were not available from this subject in later weeks). Proteobacteria blooms were present in most time points of TD, but were also found in many samples with no TD. Subject 41, who reported many cases of GI distress, had frequent blooms of Proteobacteria, but these did not appear to be linked to individual reports of GI distress. Most of the Proteobacteria detected was Gammaproteobacteria (*Escherichia-Shigella* spp.), particularly in, but not limited to, the time points of TD. Subject 34 had a bloom of Alphaproteobacteria (Rhizobiales) during the TD time point. This same taxa bloomed in non-TD time points for both subjects 24 and 25, so this taxa is not necessarily indicative of TD. There are Pasteurellales and Pseudomonadales detected in some Proteobacteria blooms, but these are not associated with TD events. The taxonomy plots can be viewed at various depths of taxonomy and with available metadata included with the QIIME2 visualization artifact (S2 File, visualize at https://view.qiime2.org/, see S2 File text).

To better understand the interactions between the gut microbiome and TD, we focused on the two subjects that had the most frequent clinical visits because of TD. Subjects, 29 and 33, had recurring (in this case, incidents that spanned more than one of our weekly samples) TD. Subject 29 is the only subject in this study to report having TD prior to enrollment, but had a negative result for the first stool submitted The subject was positive for EPEC/ETEC in the second stool submitted 13 days later, EPEC/ETEC/EAEC for the third stool submitted 30 days later, and ETEC/Cyclospora for fourth stool submitted after an additional 14 days. Subject 33 was positive for EAEC/ETEC/STEC in both stools submitted 15 days apart. The most abundant microbial families are shown by sampling week for subject 29 in Fig 3. In accordance with *Escherichia* spp. infection, there was a large overgrowth of *Enterobacteriaceae* during most of the weeks with TD incidents. Week 14 is the notable exception: during this week, a eukaryotic pathogen was detected (*Cyclospora cayetanensis*, via parasitology tests). *Bacteroidaceae*, *Rikenellaceae*, and *Ruminococcaceae* were substantially reduced during infections, while the relative abundance of *Lachnospiraceae* and *Veillonellaceae* increased. Unrelated to any apparent clinical or self-reported GI issues, there was a large bloom of Clostridiales Family XIII in week three, which was detectable but much lower in abundance during other weeks. Subject 33's bacterial families are shown in Fig 4 by sampling week. This subject experienced *Escherichia* spp. infection on weeks 15 and 17, but an *Enterobacteriaceae* bloom can be seen in the weeks preceding and following the TD events. In this subject, TD was associated with a loss of *Bacteroidaceae* and the archaea *Methanobacteriaceae*, and an increased relative abundance of *Akkermansiaceae*. Week 30 also shows a bloom of *Enterobacteriaceae*, but in this case, the surrounding weeks lack the bloom or shifts in other taxa observed for the prior TD events. There was no clinical or self-reported GI distress for this week, indicating that this bloom was not sufficient to cause debilitation to this subject. Additionally, later weeks showed an increased abundance of *Bacteroidaceae* family, while many Firmicutes and *Methanobacteriaceae* decreased, with no apparent health effects. We should be careful to note that these microbial shifts could simply be indicative of TD, rather than causative (e.g., *Akkermansiaceae* may seemingly be increased in stool abundance due to TD-related sloughing of epithelial cells).

Differences in abundances of particular taxa could indicate TD resistance or susceptibility. Using a linear mixed model, to account for the non-independence of longitudinal subject data, differentially abundant SVs were detected after false discovery rate correction, between the subjects who experienced clinically diagnosed TD versus those who did not (these calculations include all time points before and after TD for the subjects, with the specific time points of TD

## Subject 29 Fecal Taxonomy by Sampling Week

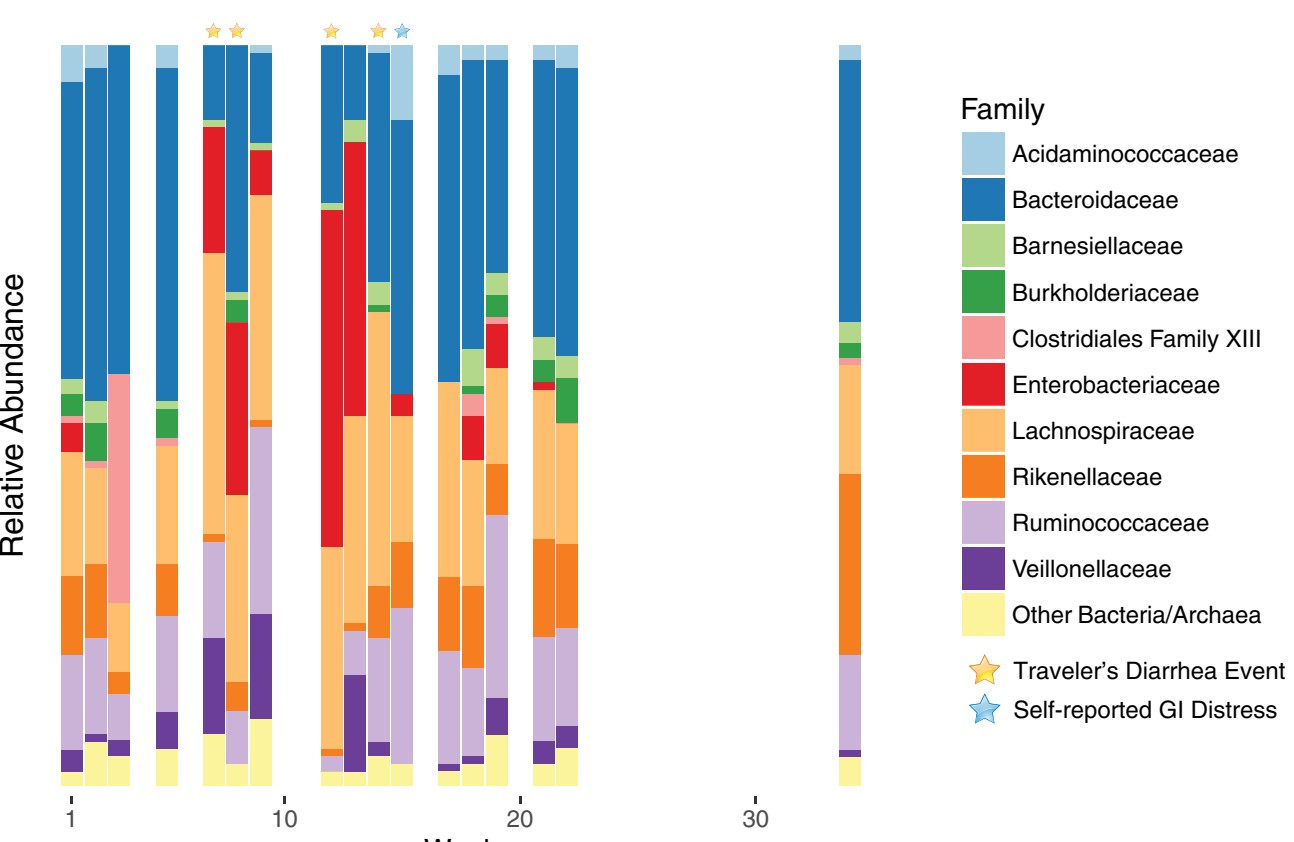

**Fig 3. Family level taxa over time for Subject 29.** Time series of family-level taxa for a subject experiencing multiple episodes of TD. Subject 29 is shown with taxa from the ten most abundant bacterial families for this subject, with the remaining bacteria and archaea collapsed into a single category.

removed). These SVs were both classified to the *Ruminococcaceae* family, with a *Ruminococcaceae* UCG-013 SV more abundant in TD+ subjects, while a *Ruminiclostridium* sp. SV had higher relative abundance in TD- subjects (Fig 5, S3 Table). These taxa may play a role in susceptibility to or protection from TD.

Using a similar approach, continuous data from self-reported GI distress (with higher values indicating more frequent logging of GI distress with the daily iPod questionnaire) were tested for correlations to the relative abundances of SVs using a linear mixed model. Two SVs, after meeting normality assumptions and FDR (Benjamini-Hochberg) correction, were shown to have a positive relationship with GI distress (S3 Table). One is a Gammaproteobacteria, a *Haemophilus* sp. (slope 1.77, FDR corrected p-value 0.0007), and the other is *Turicibacter* sp. (slope 1.57, FDR corrected p-value 0.016), in the Firmicutes phylum. We also examined self-reported stool consistency. We noticed that the Pseudomonadales and Rhizobiales bloom (in the non-TD time points) described earlier had self-reported stools that were harder than the subject's average. However, these data were only a handful of time points, and no taxa were significantly associated with stool consistency using linear mixed model testing.

## Subject 33 Fecal Taxonomy by Sampling Week

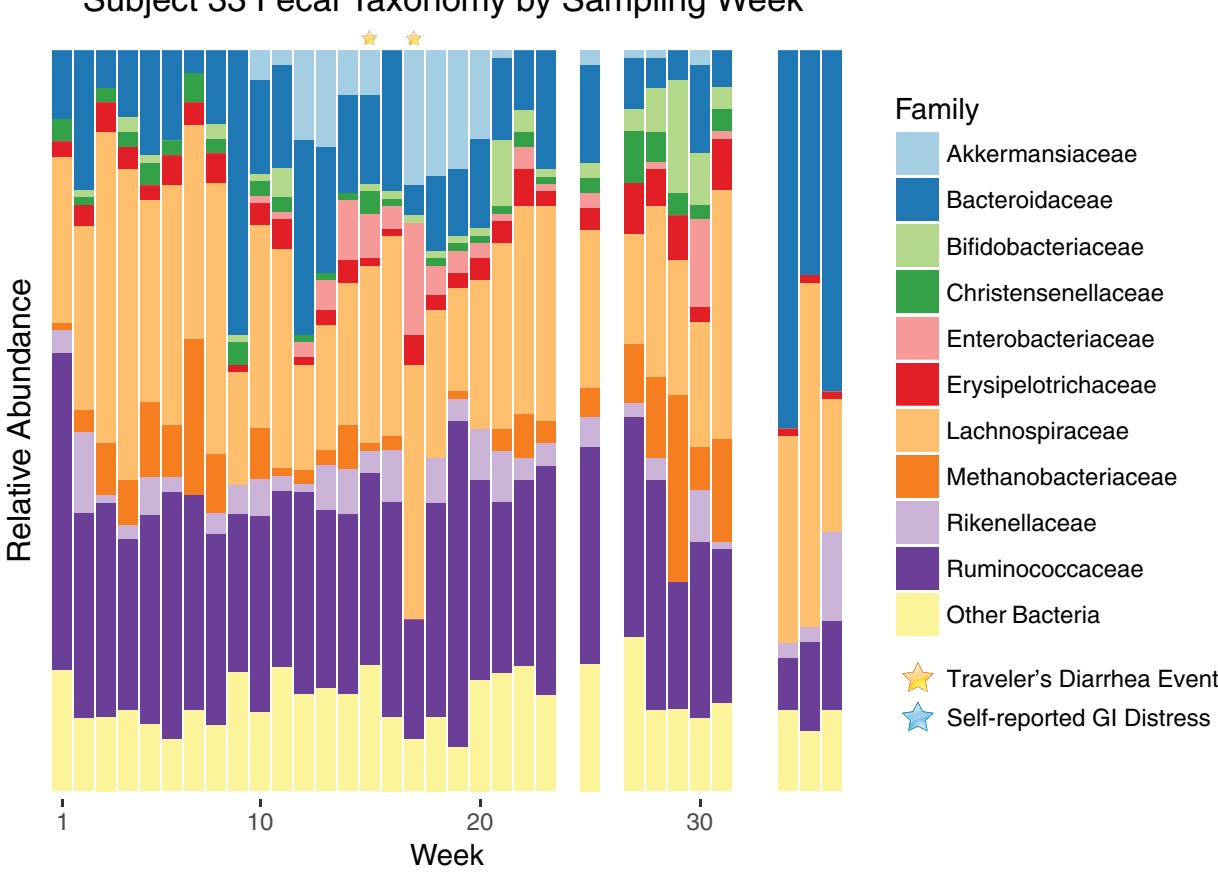

**Fig 4. Family level taxa over time for Subject 33.** Shifts in family-level taxa for subject 33, who experienced multiple episodes of TD. Taxa are shown from the ten most abundant families for this subject, with the remaining bacteria collapsed into a single category.

Alpha diversity differences between TD+ and TD- subjects were tested for via a linear mixed model as described for differential SV abundances. No differences were detected with multiple metrics. Longitudinal differences between the TD- and TD+ groups for alpha and beta diversity measures/metrics was tested with the QIIME2 longitudinal plug-in; no significant results were detected (S1 Fig).

The samples in our cohort of subjects deployed to Honduras were compared to subjects from a global fecal survey [41], which included individuals from the USA, Venezuela, and Malawi. Our cohort clustered closely to the USA adult population with a weighted UniFrac metric (Fig 6A, S3 File, see S3 File text). Interestingly, there did not appear to be any divergence over time away from this initial clustering of samples, indicating general stability in the microbiome for this group deployed from the USA for a prolonged period to Honduras (Fig 6B). Based upon self-reported diet data, these subjects mostly consumed food and water provided at the base rather than local venues, limiting potential microbial colonization and shifting of these subjects' microbiome. Subjects clustered with themselves over time (Fig 6C), as expected, and the particular time points of TD did not show a strong divergence from the subjects' other time points (Fig 6D).

We detected a higher Firmicutes:Bacteroidetes ratio in subjects who did not get TD, with an average of 2.08 Firmicutes:Bacteroidetes for TD- subjects versus 1.64 for TD+ subjects

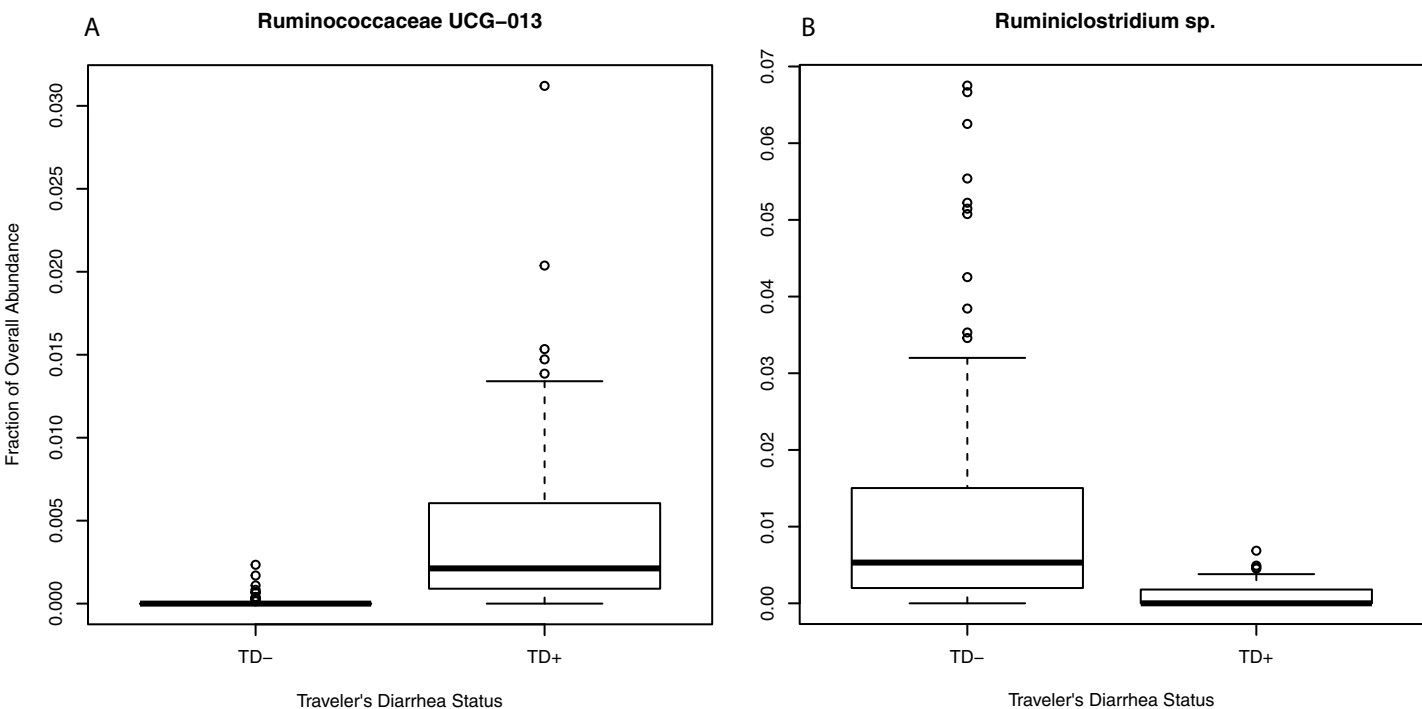

**Fig 5. SVs differentially abundant based upon traveler's diarrhea status.** Two SVs showed differential abundance with a linear mixed model approach and after FDR correction for TD+ vs TD- subjects. The relative abundance of each are shown, separated by subjects who experienced TD versus those who did not (specific time points of TD incidents are not included). The most specific taxonomic classification for the sequence variant is shown; *Ruminococcaceae* UCG-013 in A (p-value 0.0053), and a *Ruminiclostridium* spp. in B (p-value 0.040). The p-values are FDR corrected.

(Firmicutes/Bacteroidetes fractional abundance of 0.571±0.115/0.274±0.128 for TD- subjects, and 0.522±0.073/0.318±0.162 for TD+ subjects, with standard deviation shown). This is consistent with a prior observation for subjects affected by, or resistant to, TD [19]. If this pattern can be confirmed with larger sample sizes, it may suggest differential resistance, i.e., the host's microbial community occupies niches and prevent TD-causing microbes from successfully colonizing the host, or resilience to TD-causing microbes, in which case the host's microbes can be disrupted but rebound quickly and limit the effects of TD.

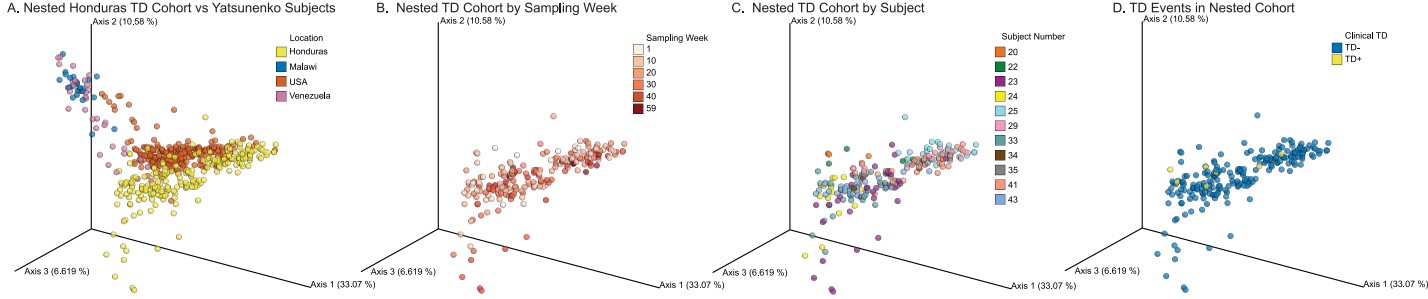

**Fig 6. PCoA of study 2b subjects deployed to Honduras.** Samples are shown in a principal coordinate plot, clustered by weighted UniFrac distances, with the axis numbers showing the percentage variation explained by that axis. A) Samples from the current study compared to samples from other locations (Yatsunenko et al [41]). Samples are colorized by geographic location. B) Samples from the study 2b are shown in a time gradient (week of sampling; week 1 indicates the first week of overseas deployment). C) All samples from each subject in study 2b are colorized by individual. D) The specific time points of TD incidents in the study 2b are highlighted. See also S2 File.

## Discussion

Utilizing questionnaire data from over 1000 subjects deployed to a forward operating base in Honduras, we confirmed that travelers' diarrhea (TD) is a persistent issue even in a location that has modern facilities and a relatively slow pace of operations. Of those people experiencing TD during their deployment, the majority reported greater than three stools per 24-hour period, resulting in reduced performance. From our passive and active surveillance studies (Studies 1 and 2a), the majority of TD cases had multiple pathogens associated with each episode, with diarrheagenic *E. coli* strains common in many of the cases. TD in deployed personnel was shown to have a complex etiology, which has implications for its prevention and treatment. It is important to note that the rate of TD was much lower in this study than rates reported in other frontline bases and could be reflective of the more modern facilities, predominantly onsite dining, and lower pace of operations compared to other locations. It is also important to note that this is the clinical incidence of TD, so there may have been cases of mild TD that were not captured. Similarly, questionnaires performed at the end of a person's deployment may not accurately capture what they actually experienced because of recall bias. Additionally, we were not able to recruit and enroll subjects prior to their arrival at the military base, so recruitment frequently occurred in the second or third week after the personnel had arrived and been processed on site. Considering these constraints, it is likely the actual incidence rate of diarrhea among this population is much higher than reported here.

The role of the gut microbiome in TD is beginning to be explored. Pop and colleagues [15] simulated TD by inoculating volunteers (from the USA) with ETEC strain H10407, and tested for differences in the microbiomes of those who acquired TD versus those who were resistant. Youmans, et al. [19] observed samples of subjects within 72 hours of TD incidents, finding that the healthy travelers (N = 12) and TD subjects (N = 99) clustered distinctly (inter-subject differences, beta diversity), but the healthy travelers had a higher Firmicutes:Bacteroidetes ratio. We also observe a higher Firmicutes:Bacteroidetes ratio in our TD- subjects relative to the TD+ individuals. Although a number of taxa were differentially abundant between the two groups (described previously), there was no difference in intra-subject diversity (alpha diversity) with healthy controls. Similarly, no intra-subject diversity (alpha diversity) differences were found in the pre-travel samples of healthy and TD subjects in a cohort of 43 individuals that were sampled before and after traveling to tropical locations worldwide [16]. Our results are consistent with these prior observations, as we detect no differences for alpha diversity between TD+ and TD- subjects. It should be emphasized that while our results are not statistically significant, the number of independent samples are small (N = 4 TD- subjects, N = 7 TD + subjects).

Microbial taxonomies associated with the subjects who acquired TD versus those that did not for these three studies are shown in S5 Table. There is modest overlap between their results, and *Prevotella copri* showed opposite effects—however, it should be noted that both the sampling and statistical methodology are different among these three studies. There could be particular strains of *P. copri* providing differential susceptibility or protection, and this group could behave differently depending upon the context of the population. For example, children with higher abundance of *P. copri* in developing countries tend to be protected from diarrhea [17]. In an extensive longitudinal study of two subjects [42], changes in the microbial community were detected during episodes of diarrheal illness, and the community maintained an altered stable state in one subject even after recovery; the long-term impacts of altered gut microbiota on host health following TD are unexplored. In *Campylobacter* spp. specific studies of TD and its relationship to the microbiome, taxa of the Firmicutes phylum were noted to be in increased relative abundance in subjects protected from infection. Kampmann and

colleagues showed enrichment of the Lachnospiraceae family (*Dorea* and *Coprococcus* genera) [43] was found in subjects that did not acquire *Campylobacter* spp.-causing TD. Dicksved et al showed that the Clostridiales order was enriched in subjects that did not become infected with *Campylobacter* spp. [44] in particular, Lachnospiraceae (unclassified) and the *Anaerovorax* genera. Lower relative abundance of the Bacteroidetes phylum, and species in the *Escherichia*, *Phascolarctobacterium*, and *Streptococcus* genera, was observed in these subjects. We did not observe TD-causing *Campylobacter* spp. in our study, and thus these prior results may not be directly applicable to our observations, however, there could be a general mechanism, such as niche occupation and resistance to depletion by particular Firmicutes for microbiome-derived TD resistance.

In this study, we detected bacteria that were differentially abundant in healthy subjects versus those that contracted TD. These were both taxa within the *Ruminococcaceae* family, and they may play a role in providing protection (*Ruminiclostridium* sp.) or susceptibility to TD (*Ruminococcaceae* UCG-013). *Ruminiclostridium* can produce short-chain fatty acids [45] that aid in epithelial integrity, which could explain its protective effect observed in this study. Limited information about the uncultured *Ruminococcaceae* UCG-013 group is available in regards to GI diseases, however, a customized diet intervention to mitigate the effects of inflammatory bowel disease caused a depletion of *Ruminococcaceae* UCG-013 [46], indicating a potential role in GI health. The few prior studies of TD showed limited overlap with our results, or each other, but *Ruminococcaceae* had taxa that both exacerbated and protected from TD [15,16]. Our results did not show differences in alpha diversity of the TD+ and TD- subjects, which replicates prior observations [16,19]. We tested the hypothesis that a microbiome that was volatile (either in overall, or alpha diversity, or shifting in membership, beta diversity) was more susceptible to TD, however, we did not detect significant differences to support this notion. Nevertheless, we did qualitatively observe disruptions to the microbiome in time points prior to actual TD events for a subject with recurrent TD. Taxa associated with self-reported GI distress were also detected (*Haemophilus* and *Turicibacter* spp.). *Haemophilus* spp. have been associated with diarrheal disease [45], while the *Turicibacter* sp. is more difficult to explain—the depletion of this taxa was detected in irritable bowel syndrome patients [46], showing a potentially contradictory role to our self-reported GI distress subjects.

Multiple studies have been conducted to assess the incidence, etiology, and immediate impact on health and military readiness due to TD [2,4,47–49]; however, much less focus has been given to possible persistent symptoms and diminished quality of life in the aftermath of these acute syndromes, which may result in a significant burden of disease among returning veterans [20]. The potential of these pathogens to cause continued sequelae beyond the duration of deployment may be difficult to detect, especially for National Guard and Army Reserve personnel who return to civilian lives and are not frequently followed by the Military Health System [21]. Post-infectious irritable bowel syndrome (PI-IBS) is one of the major concerns following GI disease in military personnel. PI-IBS is a constellation of functional gastrointestinal symptoms which occur and persist following the resolution of TD. Two recent meta-analyses showed a 7–8 fold increase in relative risk for IBS following primary infection compared with healthy controls [50,51]. This risk appears to remain elevated at 24–36 months after infection [50]. In addition, recent metagenomics have demonstrated that changes in anaerobic gut bacteria (shifts from beneficial Bacteriodetes phyla to potentially harmful species of the Firmicutes phyla, specifically those from the Clostridiales family) correlate with cases of inflammatory bowel disease and irritable bowel syndrome (including PI-IBS) relative to the gut microbial community of healthy persons [51–56]. These microbial community changes may result in increased gas production by bacteria, decreased immune tolerance, increased inflammation of the gut, and reduced gut-absorption, all of which correlate to the symptoms of IBD

and IBS [51]. Long term changes (i.e. weeks or months after a TD incident) in an individual's stool microbiome were not observed in our study after a TD episode, with their microbial community generally reverting back to baseline. However, qualitatively, the stool microbiome did exhibit change immediately prior to and post TD episode.

It should be noted that our study has constraints. The number of independent subjects enrolled in Study 2a and 2b was limited due to difficulties with recruitment in an active military operational setting, and a much larger cohort of individuals will be required to confirm our observations or to build accurate predictive models for susceptibility that are not at risk of over-fitting. Subjects volunteered to participate, and as such, this created a bias towards individuals who are willing to participate in survey and fecal sampling studies. Additionally, the results may be context dependent; the microbial signature that indicates protection or susceptibility to TD may not apply to non-US populations and, within US populations, the same gut microbial community may not be indicative of susceptibility or protection from TD if challenged by an environment with a different microbial milieu (e.g., TD-causing *Campylobacter* spp. in southeast Asia). All US military personnel deployed to areas at a high risk of malaria are required to take doxycycline as an antimalarial prophylactic. Doxycycline has also been used in the prevention of TD [57,58], which may impact TD rates in our study population. We only had access to subjects while they were deployed and hence we have no control group that were on site but not taking doxycycline. Long-term use of antibiotics [59,60] can persistently alter the gut microbiome, however, we do not have samples prior to or after deployment, and due to this cannot address the impact of doxycycline on these subjects. While this is an important issue to investigate in future studies, addressing the question of long-term effects of doxycycline goes beyond the scope of this study.

Despite these constraints, this study provides the most comprehensive longitudinal TD and microbiome analysis of personnel in a deployed, forward operating base to date. The nature of the environment ensured that subjects experienced similar conditions throughout the length of the study, including diet and food accessibility, physical activity levels, and deployment-associated stressors that would not have been easily replicated in a non-deployment scenario.

Future studies, to extend these results, should include a larger pool of independent subjects for better predictive models, metagenomic sequencing of samples to identify novel, TD-causing pathogens, and long-term studies of individuals who suffer post-TD irritable bowel syndrome or other GI disorders [61], for any potentially causative role related to microbial dysbiosis.

Collectively, with the caveat that these data are subjectively reported, the findings illustrate that even in a military base with a modern dining facility, good sanitation, and housing, TD is still problematic and impactful and the etiology of TD is complex, frequently associated with multiple pathogens. Our analyses of the gut microbiome provide tantalizing clues into its role in TD, with a *Ruminiclostridium* spp. associating with resistance and an uncultured Ruminococcaceae UCG-013 taxa associating with susceptibility to TD in this study, as well as an observation of disrupted microbiota several weeks before TD events (subject 33) which could serve as an indicator of susceptibility. Analyses of longitudinal studies and/or larger cohorts during travel and deployment are likely to identify strains and communities within the gut microbiome that provide resiliency or susceptibility to TD. These data could provide an important component to the treatment and prevention of TD, possibly through modulation of the gut microbiome using prebiotic, probiotic, or synbiotic methods.

## Supporting information

**S1 Fig. Longitudinal volatility tests of alpha and beta diversity.** QIIME2 longitudinal output is shown for observed OTUs (A), Faith's phylogenetic diversity (B), Evenness (C), Shannon

(D) and Chao1 (E) alpha diversity measures, and Jaccard (F), unweighted UniFrac (G), weighted UniFrac (H), and Bray-Curtis (I) beta diversity metrics/dissimilarity.
(EPS)

**S1 File. Post-deployment questionnaire.**
(PDF)

**S2 File. Taxonomy plots in QIIME2 artifact format.** This includes per-sample metadata, and can be viewed at https://view.qiime2.org/. For example, choose taxonomic level 3 to see the class level, and under "Sort Samples By" select "Subject", then click + to add additional sorting for "Order" and again for "ClinicalTD". Taxonomies can be toggled by clicking the colored box next to the taxa.
(QZV)

**S3 File. Weighted UniFrac PCoA plot of TD subjects versus adult subjects from Yatsunenko et al [41] in QIIME2 artifact format.** This can be viewed at https://view.qiime2.org/. For example, to view the data by TD subject, select "Subject" under the scatter dropdown box. Successive time points can be connected by clicking the animations tab, selecting Gradient->Order, Trajectory->Subject, and clicking the play button.
(QZV)

**S1 Table. iPod Touch questionnaire data.** The raw data, plus a manually entered "Corrected-Date" column, are included.
(XLSX)

**S2 Table. QIIME-compatible metadata mapping file with parsed data from the iPod questionnaire (including inferred sleep duration) included.**
(TXT)

**S3 Table. Linear mixed model results for microbial abundances versus TD category and versus self-reported GI distress levels for study 2b samples.** The anova output from lmer results, plus FDR-corrected p-values are shown.
(XLSX)

**S4 Table. Merged QIIME-compatible metadata mapping file for samples of Yatsunenko et al and study 2b samples.**
(TXT)

**S5 Table. Summary of microbial taxonomies from prior studies which distinguish human subjects that acquired TD and those that did not.**
(XLSX)

**S1 Appendix. This shows the R environment used for the linear mixed model testing.**
(TXT)

## Acknowledgments

We would like to thank Qiaojuan Shi for sample handling, and Ronda Boles for sample shipping logistics.

The views expressed in this article reflect the results of research conducted by the author and do not necessarily reflect the official policy or position of the Department of the Navy, Department of Defense, Uniformed Services University of the Health Sciences, The Henry M. Jackson Foundation for the Advancement of Military Medicine, Inc., nor the United States Government.

The authors are military service members or federal/contracted employees of the United States government, with the exception of WAW and REL. This work was prepared as part of MPS's official duties. Title 17 U.S.C. 105 provides that "copyright protection under this title is not available for any work of the United States Government." Title 17 U.S.C. 101 defines a U.S. Government work as work prepared by a military service member or employee of the U.S. Government as part of that person's official duties.

## Author Contributions

**Conceptualization:** William A. Walters, David R. Tribble, Adam P. Irvin, Ramiro L. Gutierrez, Mark S. Riddle, Ruth E. Ley, Michael S. Goodson, Mark P. Simons.

**Data curation:** William A. Walters, Faviola Reyes, Michael S. Goodson, Mark P. Simons.

**Formal analysis:** William A. Walters, Mark P. Simons.

**Funding acquisition:** David R. Tribble, Nancy Kelley-Loughnane, Michael S. Goodson, Mark P. Simons.

**Investigation:** William A. Walters, Faviola Reyes, Nathanael D. Reynolds, Jamie A. Fraser, Ricardo Aviles, Ramiro L. Gutierrez, Michael S. Goodson, Mark P. Simons.

**Methodology:** William A. Walters, Faviola Reyes, Giselle M. Soto, Adam P. Irvin, Ramiro L. Gutierrez, Ruth E. Ley, Michael S. Goodson, Mark P. Simons.

**Project administration:** Giselle M. Soto, Nancy Kelley-Loughnane, Ramiro L. Gutierrez, Ruth E. Ley, Michael S. Goodson, Mark P. Simons.

**Resources:** Faviola Reyes, Giselle M. Soto, Jamie A. Fraser, Ricardo Aviles, David R. Tribble, Ramiro L. Gutierrez, Mark S. Riddle.

**Software:** William A. Walters.

**Visualization:** William A. Walters, Michael S. Goodson, Mark P. Simons.

**Writing – original draft:** William A. Walters, Michael S. Goodson, Mark P. Simons.

**Writing – review & editing:** William A. Walters, Faviola Reyes, Adam P. Irvin, Nancy Kelley-Loughnane, Ramiro L. Gutierrez, Mark S. Riddle, Ruth E. Ley, Michael S. Goodson, Mark P. Simons.

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
