## [Decision Letter · Decision Letter 0]

9 Apr 2020

PONE-D-19-31202

Epidemiology and associated microbiota changes in deployed military personnel at high risk of traveler's diarrhea

PLOS ONE

Dear Walters,

Thank you for submitting your manuscript to PLOS ONE. After careful consideration, we feel that it has merit but does not fully meet PLOS ONE’s publication criteria as it currently stands. Therefore, we invite you to submit a revised version of the manuscript that addresses the points raised during the review process.

While all of the reviewers agreed that this is an interesting and important study, the reviewers noted some issues that need to be adequately addressed to improve the manuscript presentation, data analysis, and overall rigor and quality. 

We would appreciate receiving your revised manuscript by May 24 2020 11:59PM. To enhance the reproducibility of your results, we recommend that if applicable you deposit your laboratory protocols in protocols.io, where a protocol can be assigned its own identifier (DOI) such that it can be cited independently in the future. For instructions see: http://journals.plos.org/plosone/s/submission-guidelines#loc-laboratory-protocols

We look forward to receiving your revised manuscript.

Kind regards,

Brenda A Wilson, Ph.D.

Academic Editor

PLOS ONE

Additional Editor Comments (if provided):

While all of the reviewers agreed that this is an interesting and important study, the reviewers noted some issues that need to be adequately addressed to improve the manuscript presentation, data analysis, and overall rigor and quality.

We note that one or more of the authors are employed by a commercial company: The Henry M. Jackson Foundation for the Advancement of Military Medicine, Inc.

Reviewers' comments:

Reviewer's Responses to Questions

**Comments to the Author**

1. Is the manuscript technically sound, and do the data support the conclusions?

Reviewer #1: Partly

Reviewer #2: Yes

Reviewer #3: Yes

Reviewer #4: Partly

2. Has the statistical analysis been performed appropriately and rigorously? 

Reviewer #1: No

Reviewer #2: Yes

Reviewer #3: Yes

Reviewer #4: Yes

3. Have the authors made all data underlying the findings in their manuscript fully available?

Reviewer #1: Yes

Reviewer #2: Yes

Reviewer #3: No

Reviewer #4: Yes

4. Is the manuscript presented in an intelligible fashion and written in standard English?

Reviewer #1: Yes

Reviewer #2: Yes

Reviewer #3: Yes

Reviewer #4: Yes

5. Review Comments to the Author

Reviewer #1: Traveller’s diarrhea is a highly important and relevant topic to military personnel and civilian traveller’s alike, and there is substantial interest in determining etiology, and treatment and recovery strategies. Few studies have examined the gut microbiota as a factor in traveller’s diarrhea. As such, this study will make a welcomed and important contribution to the evidence base. I congratulate the study team for conducting what was undoubtedly a difficult and time-consuming study to execute. This likely contributed to the small sample size for microbiota analyses. This is obviously a study limitation, and is readily acknowledged by the authors. I do think there are several areas where this paper could be improved, particularly in the methods which I think could be presented more clearly given the inclusion of multiple sub-studies/cohorts in the paper, and in the analyses which could be expanded. Specific comments are below.

MAJOR COMMENTS:

1) The methods used for each study are somewhat difficult to determine (e.g., see minor comments below). Please consider separating into study 1, study 2, etc. and describing the purpose of each study, the methods used for each study, and the results for each study separately. A figure to show the different cohorts, the samples they provided, and when those samples/data were provided may be helpful. Additionally, using the same terminology to describe each study throughout the text would be helpful. Currently, it seems that active/passive surveillance, epidemiology, longitudinal, etc. are used variably and sometimes interchangeably throughout creating confusion as to which methods and results correspond to which cohort. Finally, sample size calculations are needed, or, if none, the rationale for the sample sizes within each study should be clarified.

2) Much of the results are presented as descriptive analyses, including those generated from the most novel cohort in this study, the n=11 longitudinal cohort (e.g., Ln 412-471). To increase the impact of this important research, the authors are encouraged to consider additional analyses. For example, were there any demographic or behavioral predictors of TD in the retrospective or n=67 longitudinal study? In the n=11 cohort, considering questions such as differentially abundant taxa prior to and after TD development (in addition to throughout) in TD+ vs TD- subjects, and determining whether pre-TD composition was restored in TD+ subjects following TD would be of interest (e.g., Pop M, et al. BMC Genomics 2016;17:440; Dethlefsen and Relman, PNAS 2011;108:4554). Given the small n, perhaps using an n of 1 type approach (e.g., Vohra S et al., BMJ 2015;350:h1738) would be useful. Statements such as those made in ln 624-626 lead one to believe that some of this analysis has already been completed. If so, please consider whether those data can be more clearly identified and/or incorporated into the results.

MINOR COMMENTS:

2) Ln 123-126: More details regarding the methods used for this cohort is needed. For example, was data extracted from medical records, if so, what data? Were stool samples collected?

3) Ln 120-121: Suggest removing anecdotes. Shouldn’t studies 1 and 2 provide data to estimate TD rates? Are there any historical data to support the statement?

4) Ln 126-131: Please clarify if these questionnaires have been validated and/or if other published studies have used the same questions to assess incidence of TD. Given the retrospective design and that TD is more likely early in deployment, how reliable can we expect these data to be? This should be considered in the study limitations.

5) How were TD and GI distress defined in the different study cohorts?

6) Please include inclusion/exclusion criteria, if any, for the n=67 and n=11 cohorts.

7) Ln 146: How were these subjects selected? If not random, this could bias results and should be considered as a limitation.

8) Ln 158: Passive or active surveillance/ study 1 or study 2?

9) Ln 159-162: What groups are being compared?

10) Ln 169: More detail on the univariate analyses used is needed (e.g., type of regression, variables were included in the model, etc.). Ln 310-362 report descriptive analyses, but where are the analyses described in ln 169-172 presented?

11) Ln 174, 208: Details regarding stool sample collection methods (e.g., who, where, how, when) are needed.

12) Ln 323: Please define “reduced performance”

13) Ln 474-481: Please clarify whether these SVs were differentially abundant before or after TD events or both?

14) Ln 533: The meaning of “tolerance of disrupted peristalsis” is unclear, please clarify.

15) Ln 550-552: Were the n=11 subjects queried as to whether they had already experienced TD prior to study enrollment? If so, this should be considered in the analysis. If not, this should be discussed as a limitation.

16) Ln 573-574: Did the authors also consider including: Kampmann C, et al. Clin Microbiol Infect 2016;22:61 or Dicksved J et al. mBio 2014;5:e01212-14?

17) Ln 645-646: Im not sure that the results or discussion support this statement. Suggest removing.

18) Suggest avoiding military terminology as much as possible (e.g., operational tempo, forward operating base, etc.).

19) Figure 6: This appears to be the first place Honduras, presumably the study site, is mentioned. Please clarify that the Honduras subjects are from the present study, and none of the Honduras data points are from the Yatsuneko study.

Reviewer #2: The authors present an interesting study to understand the effect of traveler’s diarrhea on deployed military subjects, including incident rate, identification of pathogens and changes in gut microbiome. Literature about the effects of TD on deployed personnel with regard to microbiome is limited. Results from this study begins to identify how the microbiome both effects TD and is affected by TD. This paper highlights the difficulties of conducting this type of research. Study constraints and challenges were discussed, as well as next steps to extend the results.

1. Content: Comments made to illustrate points that are unclear or changes to address document flow/readability

Line 353-354 Including the reason for the differences in Shigella detection using plating vs non-culture method to illustrate the limitation of culture based methods would be informative.

Line 441 Transition sentence explaining reason for examining subjects 29 and 33 by sampling week would be helpful and improve document flow

Line 475 Differential SV detection was presented. State the reason for examining SVs.

Lines 496-500 Sentence long and a difficult read. Break into separate sentences (stool consistency and time points)?

Line 507 Specify the cohort being compared to the globe questionnaire is from Honduras

Lines 526-529 Is this determined from data in fig 2?

Meaning of data in parenthesis is unclear in:

• Table 1 Mean age values unclear (e.g., what do “(32.3, 34.7)” for females). Also on line 311

• Table 2 Diarrhea, median episode # per person: 2 (1, 40)

Days experiencing diarrhea, median per person range: 3 (1, 60)

• Table 5 Mean are range: 42.3 (36.9, 47.6), 37.6 (34.7, 40.5)

Tables 4, 6 Identifying EPEC, EAEC, ETEC, STEC in legend would be helpful to not have to refer back to the text (unless not a PLOS One convention)

Figure 5 Include p-values (in legend or in text)

Figure 6 Difficult to follow. Rewrite to clarify such that the figure and legend stands on its own and referring back to the text is not necessary.

• Legend should be reformatted to have the letter 1st, then the test (e.g.: B. US subjects deployed…)

• Panel C: State that subjects are for all time points

Figure 1 Legend needs more detail describing the bottom panel with the dots showing interactions

2. Formatting

Materials and Methods: spacing after colons in subheadings not consistent (e.g., p. 10, lines 169 and 174)

Units not consistent: h vs hour (various locations in document)

Line 98 Change followed to following

Line 286 Correct ipod to iPod

Line 266 clarify anova( )

Line 288 FDR not defined

Line 628 Correct “and- - a”

Line 183 Identification using API-20E strips mentioned, but no data presented

Lines 286-288 Rewrite to correct grammar/tense

Line 386 Is separate 5b legend here? Also in table 5 legend on line 384

Line 540 Define lower operational tempo (move from line 548)

Line 556-567 Format citations consistently. Add citation #s individually?

Reviewer #3: This manuscript discuss the epidemiology and changes in the associated microbiota in deployed military personnel at high risk of traveler's diarrhea. The information presented is appropriate to researchers interested in studying the alteration in microbiome as a causative agent for some of the prevalent disease. This topic is worthy of study and this paper makes a great start on this. The methods, data interpretation, and conclusions are appropriate. The authors themselves list the limitations of the experiments presented in the conclusion section. There were only a limited number of independent subjects enrolled in addition to this result may not apply to non-US populations due to the high variability in microbiome composition as a result of different geographic location. Acknowledging these shortcomings, the reviewer believes the manuscript stands on its own. The reviewer would like for future experiments to include the above-mentioned limitations of the present study, although this would have to be in a separate manuscript.

I just wander did the authors evaluate the effect of doxycycline as antimalarial prophylactic dose on the microbiota composition of all subjects included in this study. I think this may be responsible for some of the microbiome alteration. Finally, did the authors used negative and positive sequencing controls? It is now clearly established that reagents used in microbiome studies have readily detectable bacterial DNA (Salter BMC Biology 2014). Therefore, it is crucial to sequence reagents (and ideally, have controls at every step of samples acquisition, preservation and preparation) to control for potential contaminations

Reviewer #4: April 7, 2020

Editor-in-Chief,

PLOS ONE

I have carefully reviewed the manuscript titled "Epidemiology and associated microbiota changes in deployed military personnel at high risk of traveler’s diarrhea” authored by Walters et al. This article is aligned with the scope of PLOS ONE Journal and provide valuable information on the relationship between long-term deployment and traveler’s diarrhea (TD). TD has a significant impact on military personals and this manuscript will help guide future research in identifying potential causes of TD. To resolve underreported TD cases, authors followed a comprehensive surveillance plan among deployed military personals and to understand the etiology of TD author utilized state of art methods in conducting the microbiome analysis. However, this manuscript needs several minor revisions. Following are the concerns that need to be addressed:

Thank you

Sincerely

Akemi Wijayabahu

COMMENTS TO THE AUTHOR

General Comments:

I commend the authors for conducting a comprehensive fecal analysis for various pathogens and not limiting to gut bacteria. Also, for providing a detailed methods section. To resolve underreported TD cases, authors followed a comprehensive surveillance plan among deployed military personals and to understand the etiology of TD author utilized state of art methods in conducting the microbiome analysis.

My suggestions are to have more group-based data in the results section and to minimize the focus on case by case data. Perhaps, the authors can include case reports in the supplemental section. Additionally, the discussion section should be more focused on TD and how different bacterial taxa relates to TD. Use of Doxycycline during the deployment period is a major factor that needs to be discussed further by relating to its impact on TD, known evidence on the dysbiosis of gut microbiota, risk of resistant infections, the protective effect of Doxy on TD risk, etc.

Reference: Diptyanusa, A., Ngamprasertchai, T., & Piyaphanee, W. (2018). A review of antibiotic prophylaxis for traveler’s diarrhea: past to present. Tropical diseases travel medicine and vaccines, 4(1), 1-8.

Minor Comments:

Abstract-

Page 2, line 32 – suggest revising the methods section to include a sentence describing the microbiome analysis. I also suggest revising the structure of the abstract to separate the purpose, methods and results sections.

Page 3, lines 54-56- The authors described the impact of TD on military operations/performance but did not provide any results related to the topic. Suggest providing some evidence (Ex- duty days lost).

“These findings illustrate the complex etiology of diarrhea amongst military personnel in deployed settings and its impacts on performance.”

Background-

Page 5, lines 91-93 (and page 4-5 paragraph)- It would be better if the authors can describe known evidence and research gap related to TD, then provide a specific hypothesis. In this paragraph, the author provides a good strong background on microbiota, function, and relationship to diseases/disorders in general. This paragraph needs at least some information relating to microbiota and TD.

“Thus, we hypothesize that gut microbial community composition can be a significant factor in symptomatic and sub-clinical syndromes.” Relating to TD or in general any illness?

Page 5, line 100- When authors mention of “illness”, does this mean any disease/disorder during deployment or just TD?

Methods-

Page 6, line 110- Maybe it would be better to include a definition for “forward deployment” within brackets/parenthesis or use a more general term (Ex- deployed to a US military base on foreign soil/Central America).

Page 6, lines 128-131- Did the questionnaire include a specific time duration? As an example, past events including signs and symptoms of diarrhea, respiratory, and febrile illnesses………during the past 30 days or 12 months? Also, I recommend including the questionnaire in the supplemental section (or provide the referenced Table SX).

Page 8, lines 160-161- I apologize for my unfamiliarity with the word “time in theater”, is this a specific room as a surgical theater? Please clarify?

Did the authors have the following baseline information? time since deployment, history of prior diarrheal infections, race, risk factors contributing to the transmission of pathogenic bacteria within the deployment site (hygienic behavior assessment, availability of resources, etc.), use of other medication (laxatives), use of probiotics (yogurt)?

Page 8, lines 165-166- How did the authors treat multi-pathogen infections (co-infections) when calculating incidence for each pathogen?

Page 9, lines 207-209- Did the authors use any kind of buffer to store the fecal samples (RNAlater)?

Page 10, lines 221-227- Suggest adding a sub-heading

Page 11, lines 239-241- Suggest including a Chao1 graph to the results section if possible

Results-

Page 15, Table 2- suggest specifying the duration of the assessment (Ex- during the past 12 months?)

Page 17, Table 3- Earlier when describing other illness, it was not clear that the symptoms were associated with diarrhea or due to unrelated illnesses. Please clarify this in the methods section/other sections.

Page 19-20, Table 4- Please provide a footnote with abbreviations and definitions

Pages 23-24- Suggest including heat maps, diversity plots and figure representing relative abundance by phyla for groups (TD+ vs TD- summarized across all participants)

Discussion-

Page 30, lines 588-591- Suggest revising the sentence as I don’t see the relevance of IBD and TD the way it is presented here.

Page 31, lines 594-597- Is it possible that the changes in the gut microbiota due to doxycycline and other antimalarial prophylactics might contribute to the risk of TD? Please include past research evidence on this subject. It is possible participants taking such medication already have a disrupted gut microbiota.

Also, please briefly discuss the strengths of the study in the discussion section (controlled diet, similar physical activity levels, etc.).

Conclusion- Suggest revising the conclusion to include a summary of findings across all participants with TD compared to those without TD (or before and after TD results of individuals collectively).

6. PLOS authors have the option to publish the peer review history of their article (what does this mean?). If published, this will include your full peer review and any attached files.

Reviewer #1: No

Reviewer #2: No

Reviewer #3: No

Reviewer #4: Yes: Akemi T Wijayabahu

---

## [Author Response · Author response to Decision Letter 0]

21 May 2020

Dear editor and reviewers,

We thank the editor and reviewers for their time and comments. Our responses to the issues raised by the editor and the reviewers are listed immediately below for our responses to questions raised by the editor, and following each question for the reviewers. Our responses are below, following the questions raised by the editor and reviewers.

Subject: PLOS ONE Decision: Revision required [PONE-D-19-31202] - [EMID:fdfe19777cc69a83]

Date: 9 Apr 2020 07:19:35 -0400

From: PLOS ONE <em@editorialmanager.com>

Reply-To: PLOS ONE <plosone@plos.org>

To: William A. Walters <william.walters@tuebingen.mpg.de>

PONE-D-19-31202

Epidemiology and associated microbiota changes in deployed military personnel at high risk of traveler's diarrhea

PLOS ONE

Dear Walters,

Thank you for submitting your manuscript to PLOS ONE. After careful consideration, we feel that it has merit but does not fully meet PLOS ONE’s publication criteria as it currently stands. Therefore, we invite you to submit a revised version of the manuscript that addresses the points raised during the review process.

While all of the reviewers agreed that this is an interesting and important study, the reviewers noted some issues that need to be adequately addressed to improve the manuscript presentation, data analysis, and overall rigor and quality.

We would appreciate receiving your revised manuscript by May 24 2020 11:59PM. To enhance the reproducibility of your results, we recommend that if applicable you deposit your laboratory protocols in protocols.io, where a protocol can be assigned its own identifier (DOI) such that it can be cited independently in the future. For instructions see: http://journals.plos.org/plosone/s/submission-guidelines#loc-laboratory-protocols

• A rebuttal letter that responds to each point raised by the academic editor and reviewer(s). This letter should be uploaded as separate file and labeled 'Response to Reviewers'.

• A marked-up copy of your manuscript that highlights changes made to the original version. This file should be uploaded as separate file and labeled 'Revised Manuscript with Track Changes'.

• An unmarked version of your revised paper without tracked changes. This file should be uploaded as separate file and labeled 'Manuscript'.

We look forward to receiving your revised manuscript.

Kind regards,

Brenda A Wilson, Ph.D.

Academic Editor

PLOS ONE

We checked the requirements from this URL (https://journals.plos.org/plosone/s/submission-guidelines) as the above URLs appear to be nonfunctional, at least at the time that we were working on the revisions. I did specifically notice that we had exceeded the 300 word limit for the abstract (https://journals.plos.org/plosone/s/submission-guidelines#loc-abstract) and we have made suggested modifications in the submitted revision

We have confirmed that the corresponding author, Michael Goodson, has a valid ORCID (0000-0002-5004-551X).

We have made sure that the samples are now publicly (previously private) available on the European Nucleotide Archive under project PRJEB31759.

Additional Editor Comments (if provided):

While all of the reviewers agreed that this is an interesting and important study, the reviewers noted some issues that need to be adequately addressed to improve the manuscript presentation, data analysis, and overall rigor and quality.

We note that one or more of the authors are employed by a commercial company: The Henry M. Jackson Foundation for the Advancement of Military Medicine, Inc.

We have included an updated section regarding the Henry M. Jackson foundation. We do note that this foundation is a non-profit institution rather than a commercial enterprise, but in any case, did not have a role in this study's design, research, or manuscript preparation.

Reviewers' comments:

Reviewer's Responses to Questions

Comments to the Author

1. Is the manuscript technically sound, and do the data support the conclusions?

Reviewer #1: Partly

Reviewer #2: Yes

Reviewer #3: Yes

Reviewer #4: Partly

2. Has the statistical analysis been performed appropriately and rigorously?

Reviewer #1: No

Reviewer #2: Yes

Reviewer #3: Yes

Reviewer #4: Yes

3. Have the authors made all data underlying the findings in their manuscript fully available?

Reviewer #1: Yes

Reviewer #2: Yes

Reviewer #3: No

Reviewer #4: Yes

4. Is the manuscript presented in an intelligible fashion and written in standard English?

Reviewer #1: Yes

Reviewer #2: Yes

Reviewer #3: Yes

Reviewer #4: Yes

5. Review Comments to the Author

Reviewer #1: Traveller’s diarrhea is a highly important and relevant topic to military personnel and civilian traveller’s alike, and there is substantial interest in determining etiology, and treatment and recovery strategies. Few studies have examined the gut microbiota as a factor in traveller’s diarrhea. As such, this study will make a welcomed and important contribution to the evidence base. I congratulate the study team for conducting what was undoubtedly a difficult and time-consuming study to execute. This likely contributed to the small sample size for microbiota analyses. This is obviously a study limitation, and is readily acknowledged by the authors. I do think there are several areas where this paper could be improved, particularly in the methods which I think could be presented more clearly given the inclusion of multiple sub-studies/cohorts in the paper, and in the analyses which could be expanded. Specific comments are below.

MAJOR COMMENTS:

1) The methods used for each study are somewhat difficult to determine (e.g., see minor comments below). Please consider separating into study 1, study 2, etc. and describing the purpose of each study, the methods used for each study, and the results for each study separately. A figure to show the different cohorts, the samples they provided, and when those samples/data were provided may be helpful. Additionally, using the same terminology to describe each study throughout the text would be helpful. Currently, it seems that active/passive surveillance, epidemiology, longitudinal, etc. are used variably and sometimes interchangeably throughout creating confusion as to which methods and results correspond to which cohort. Finally, sample size calculations are needed, or, if none, the rationale for the sample sizes within each study should be clarified.

We have endeavored to clarify the studies in the methods. The prior passive surveillance study is now listed as study 1, and the active, follow-up study, is referred to as study 2a. The nested cohort of study 2a that provided weekly fecal samples is now study 2b. We do not feel that a figure showing the different cohorts would be beneficial since subjects in Study 1 and Study 2a only provided samples if they were ill. Subjects for the microbiome longitudinal study did provide samples and answered the e-survey on a schedule, and this is shown in Fig. 2. We have worked to clarify these details in the text to give an overview of the separate studies involved.

2) Much of the results are presented as descriptive analyses, including those generated from the most novel cohort in this study, the n=11 longitudinal cohort (e.g., Ln 412-471). To increase the impact of this important research, the authors are encouraged to consider additional analyses. For example, were there any demographic or behavioral predictors of TD in the retrospective or n=67 longitudinal study? In the n=11 cohort, considering questions such as differentially abundant taxa prior to and after TD development (in addition to throughout) in TD+ vs TD- subjects, and determining whether pre-TD composition was restored in TD+ subjects following TD would be of interest (e.g., Pop M, et al. BMC Genomics 2016;17:440; Dethlefsen and Relman, PNAS 2011;108:4554). Given the small n, perhaps using an n of 1 type approach (e.g., Vohra S et al., BMJ 2015;350:h1738) would be useful. Statements such as those made in ln 624-626 lead one to believe that some of this analysis has already been completed. If so, please consider whether those data can be more clearly identified and/or incorporated into the results.

The reviewer is correct that a permanent disruption (i.e. an alternative stable state assumed after traveler’s diarrhea disruption) or resilience in the gut microbiome, where the community returned to a prior configuration, would be of interest, particularly due to a possible explanation for long-term post-deployment IBS. Unfortunately, as you have pointed out, our ability to go beyond a descriptive analysis is limited due to the number of subjects with both multiple TD events and long-term sampling. We have noted this in the text, along with the citations regarding perturbations due to antibiotic usage.

MINOR COMMENTS:

2) Ln 123-126: More details regarding the methods used for this cohort is needed. For example, was data extracted from medical records, if so, what data? Were stool samples collected?

We have modified the methods section to clarify the cohorts sampled in this study.

3) Ln 120-121: Suggest removing anecdotes. Shouldn’t studies 1 and 2 provide data to estimate TD rates? Are there any historical data to support the statement?

We have updated the text to remove the anecdote, and have provided historical context from military studies in central/south American countries. Unfortunately, historical data for this particular site in Honduras is not available, so this study is, as the reviewer points out, the estimate for TD rates at the Honduras site.

4) Ln 126-131: Please clarify if these questionnaires have been validated and/or if other published studies have used the same questions to assess incidence of TD. Given the retrospective design and that TD is more likely early in deployment, how reliable can we expect these data to be? This should be considered in the study limitations.

We do agree that self-recall can have inaccuracies, and have added text to reflect this, and included the specific questionnaire as a supplemental pdf file. We’ve also noted a prior study that has been carried out using a recall approach and a similar questionnaire. We do think that we are capturing more cases than formal TD testing detects, as many cases are mild and unreported.

5) How were TD and GI distress defined in the different study cohorts?

We have included definition for TD, and added that GI distress was listed as one of the options on the daily e-survey and was therefore self-reported.

6) Please include inclusion/exclusion criteria, if any, for the n=67 and n=11 cohorts.

We have included that subjects were recruited as soon as they reported to the medical facility at the start of their deployment. If subjects volunteered for the study, the only exclusion criteria was a stay on base less than 1 month.

7) Ln 146: How were these subjects selected? If not random, this could bias results and should be considered as a limitation.

There was no random selection. Prospective subjects were briefed on the study and contacted the study coordinator individually if they wished to participate. There was mechanisms in place to prevent coercion. For passive surveillance, individuals were enrolled when they arrived to the clinic and consented to participate. Surveys were given at the end of deployment and were voluntarily submitted. The voluntary nature of the study has been included in the text.

8) Ln 158: Passive or active surveillance/ study 1 or study 2?

We have modified the methods section to clarify the cohorts sampled in this study.

9) Ln 159-162: What groups are being compared?

We have changed the wording to more accurately describe the methods.

10) Ln 169: More detail on the univariate analyses used is needed (e.g., type of regression, variables were included in the model, etc.). Ln 310-362 report descriptive analyses, but where are the analyses described in ln 169-172 presented?

We apologize, this was from an earlier iteration of the manuscript. We have decided to modify this, and the results, section for clarity. We did not analyze beyond descriptive statistics for risk factors from the post deployment surveys as the sample sizes of the cohorts were small.

11) Ln 174, 208: Details regarding stool sample collection methods (e.g., who, where, how, when) are needed.

We have updated our methods to include the specific kit/manufacturer involved and methods for sampling/processing.

12) Ln 323: Please define “reduced performance”

‘Reduced performance’ redefined as ‘reduction in their ability to perform their job’.

13) Ln 474-481: Please clarify whether these SVs were differentially abundant before or after TD events or both?

We have modified a sentence to clarify that these are all time points except for the specific time points of traveler's diarrhea.

14) Ln 533: The meaning of “tolerance of disrupted peristalsis” is unclear, please clarify.

We have modified this statement for clarity-we wrote this as two distinct categories when the latter portion were examples of resistance and resilience.

15) Ln 550-552: Were the n=11 subjects queried as to whether they had already experienced TD prior to study enrollment? If so, this should be considered in the analysis. If not, this should be discussed as a limitation.

For the active surveillance (study 2), subjects were queried about prior diarrhea or vomiting (for the prior two weeks before enrollment). Only one subject stated that they had diarrhea, subject 29. We have modified the text to account for this.

16) Ln 573-574: Did the authors also consider including: Kampmann C, et al. Clin Microbiol Infect 2016;22:61 or Dicksved J et al. mBio 2014;5:e01212-14?

Campylobacter spp. were not detected as TD-causing agents in our study, but they are significant contributors to the disease, particularly in southeast Asia. We've added these results to the discussion with these caveats.

17) Ln 645-646: Im not sure that the results or discussion support this statement. Suggest removing.

We have removed this statement, and emphasized that further study is required to determine the potential of modulating the microbiome to effect TD.

18) Suggest avoiding military terminology as much as possible (e.g., operational tempo, forward operating base, etc.).

We take the reviewer’s point and have tried to do so as much as possible. Some military terminology is unavoidable because of the nature of the subject’s environment.

19) Figure 6: This appears to be the first place Honduras, presumably the study site, is mentioned. Please clarify that the Honduras subjects are from the present study, and none of the Honduras data points are from the Yatsuneko study.

We have modified the manuscript to emphasize that the subjects are US warfighters deployed to Honduras. We have also changed the legend of Figure 6 to clarify which data are from the current study.

Reviewer #2: The authors present an interesting study to understand the effect of traveler’s diarrhea on deployed military subjects, including incident rate, identification of pathogens and changes in gut microbiome. Literature about the effects of TD on deployed personnel with regard to microbiome is limited. Results from this study begins to identify how the microbiome both effects TD and is affected by TD. This paper highlights the difficulties of conducting this type of research. Study constraints and challenges were discussed, as well as next steps to extend the results.

1. Content: Comments made to illustrate points that are unclear or changes to address document flow/readability

Line 353-354 Including the reason for the differences in Shigella detection using plating vs non-culture method to illustrate the limitation of culture based methods would be informative.

We have added a sentence to address this comment.

Line 441 Transition sentence explaining reason for examining subjects 29 and 33 by sampling week would be helpful and improve document flow

We have added a sentence clarifying why we focused on subject 29 and 33.

Line 475 Differential SV detection was presented. State the reason for examining SVs.

We've added a sentence to address this.

Lines 496-500 Sentence long and a difficult read. Break into separate sentences (stool consistency and time points)?

We separated this sentence and modified it to improve flow.

Line 507 Specify the cohort being compared to the globe questionnaire is from Honduras

We have changed the text to highlight this.

Lines 526-529 Is this determined from data in fig 2?

Yes, and reference to Fig. 2 added.

Meaning of data in parenthesis is unclear in:

• Table 1 Mean age values unclear (e.g., what do “(32.3, 34.7)” for females). Also on line 311

This is the a 95% confidence interval, the table text was modified to reflect this.

• Table 2 Diarrhea, median episode # per person: 2 (1, 40)

Days experiencing diarrhea, median per person range: 3 (1, 60)

These are the lowest and highest values, respectively and we have changed the table to ‘min, max’.

• Table 5 Mean are range: 42.3 (36.9, 47.6), 37.6 (34.7, 40.5)

We have clarified that these are 95% CI in the text.

Tables 4, 6 Identifying EPEC, EAEC, ETEC, STEC in legend would be helpful to not have to refer back to the text (unless not a PLOS One convention)

We have added the abbreviation descriptions below Tables 4 and 6.

Figure 5 Include p-values (in legend or in text)

We have added the FDR-corrected p-values to the figure text.

Figure 6 Difficult to follow. Rewrite to clarify such that the figure and legend stands on its own and referring back to the text is not necessary.

• Legend should be reformatted to have the letter 1st, then the test (e.g.: B. US subjects deployed…)

• Panel C: State that subjects are for all time points

Figure 6 legend has been modified to incorporate the reviewer’s comments and to improve clarity.

Figure 1 Legend needs more detail describing the bottom panel with the dots showing interactions

We have modified both the figure and the legend texts (1 and 6) to address the issues with clarity raised here.

2. Formatting

Materials and Methods: spacing after colons in subheadings not consistent (e.g., p. 10, lines 169 and 174)

We have made the subheadings consistent.

Units not consistent: h vs hour (various locations in document)

We changed all reference to hours to ‘h’ 

Line 98 Change followed to following

We feel this would change the meaning of the sentence, so we modified the sentence instead.

Line 286 Correct ipod to iPod

Corrected.

Line 266 clarify anova( )

We have expanded the sentence for clarity.

Line 288 FDR not defined

We have added 'Benjamini-Hochberg' to indicate the type of correction.

Line 628 Correct “and- - a”

Corrected.

Line 183 Identification using API-20E strips mentioned, but no data presented

We have modified the text to indicate that the strips were used to identify Salmonella and Shigella from bacterial cultures as indicated in the Results section and in Tables 4 and 6.

Lines 286-288 Rewrite to correct grammar/tense

This explains the linear mixed model description and we feel changing the ‘W’ to lower case clarifies this, as it is used in line 261 of the original submission.

Line 386 Is separate 5b legend here? Also in table 5 legend on line 384

Yes, that was an oversight on our part. We have deleted the legend originally on Line 386.

Line 540 Define lower operational tempo (move from line 548)

This is a military term. We have attempted to clarify this in the text.

Line 556-567 Format citations consistently. Add citation #s individually?

We have altered this section to have consistent citations with the rest of the document, and separated the citations to be part of the individual relevant sentences.

Reviewer #3: This manuscript discuss the epidemiology and changes in the associated microbiota in deployed military personnel at high risk of traveler's diarrhea. The information presented is appropriate to researchers interested in studying the alteration in microbiome as a causative agent for some of the prevalent disease. This topic is worthy of study and this paper makes a great start on this. The methods, data interpretation, and conclusions are appropriate. The authors themselves list the limitations of the experiments presented in the conclusion section. There were only a limited number of independent subjects enrolled in addition to this result may not apply to non-US populations due to the high variability in microbiome composition as a result of different geographic location. Acknowledging these shortcomings, the reviewer believes the manuscript stands on its own. The reviewer would like for future experiments to include the above-mentioned limitations of the present study, although this would have to be in a separate manuscript.

I just wander did the authors evaluate the effect of doxycycline as antimalarial prophylactic dose on the microbiota composition of all subjects included in this study. I think this may be responsible for some of the microbiome alteration.

No, we did not. We take the reviewer's point, but all personnel are required to take doxycycline as a prophylactic antimalarial. We did not have the ability to have a control group that did not take doxycycline and have added statements to address this. We do think this would make an excellent follow up study.

Finally, did the authors used negative and positive sequencing controls? It is now clearly established that reagents used in microbiome studies have readily detectable bacterial DNA (Salter BMC Biology 2014). Therefore, it is crucial to sequence reagents (and ideally, have controls at every step of samples acquisition, preservation and preparation) to control for potential contaminations

The reviewer has raised a good point, and we have added comments in the methods section regarding negative and positive controls.

Reviewer #4: April 7, 2020

Editor-in-Chief,

PLOS ONE

I have carefully reviewed the manuscript titled "Epidemiology and associated microbiota changes in deployed military personnel at high risk of traveler’s diarrhea” authored by Walters et al. This article is aligned with the scope of PLOS ONE Journal and provide valuable information on the relationship between long-term deployment and traveler’s diarrhea (TD). TD has a significant impact on military personals and this manuscript will help guide future research in identifying potential causes of TD. To resolve underreported TD cases, authors followed a comprehensive surveillance plan among deployed military personals and to understand the etiology of TD author utilized state of art methods in conducting the microbiome analysis. However, this manuscript needs several minor revisions. Following are the concerns that need to be addressed:

Thank you

Sincerely

Akemi Wijayabahu

COMMENTS TO THE AUTHOR

General Comments:

I commend the authors for conducting a comprehensive fecal analysis for various pathogens and not limiting to gut bacteria. Also, for providing a detailed methods section. To resolve underreported TD cases, authors followed a comprehensive surveillance plan among deployed military personals and to understand the etiology of TD author utilized state of art methods in conducting the microbiome analysis.

My suggestions are to have more group-based data in the results section and to minimize the focus on case by case data. Perhaps, the authors can include case reports in the supplemental section.

We thank the reviewer/editor for their comments. We wanted to strike a balance in our analyses of the data: on the one hand we attempted to analyze grouped data for microbial markers of susceptibility or resilience to TD; and on the other hand, focus on those subjects that experienced multiple TD episodes to look for microbial differences in abundance before and after, as well as to see if there were any indicators of TD prior to it. As we noted, there were some shortcomings in the data collection which meant that these data were not available for every subject. We believe we have presented data that get at the original questions we were posing. However, we are very proud of our data set and have provided links and tools for the readers to explore the data as they see fit. As you can imagine, with the nature of the subjects in this study, there are certain restrictions on which data we are allowed to publicly release, enforced by our IRB and the Unit Commanders. We have made available everything we have been allowed to.

Additionally, the discussion section should be more focused on TD and how different bacterial taxa relates to TD.

We have expanded our discussion to incorporate the reviewer’s suggestion.

Use of Doxycycline during the deployment period is a major factor that needs to be discussed further by relating to its impact on TD, known evidence on the dysbiosis of gut microbiota, risk of resistant infections, the protective effect of Doxy on TD risk, etc.

Reference: Diptyanusa, A., Ngamprasertchai, T., & Piyaphanee, W. (2018). A review of antibiotic prophylaxis for traveler’s diarrhea: past to present. Tropical diseases travel medicine and vaccines, 4(1), 1-8.

Our sampling did not allow us to determine if doxycycline affected the subject’s microbiome or their susceptibility to diarrhea, since all personnel are required to take doxycycline as a prophylactic antimalarial when deployed to locations where the risk of malaria is high. We have expanded our discussion to address the reviewer’s points, but do not feel that we can comment on any risks associated with prophylactic doxycycline administration since it is outside the scope of this study.

Minor Comments:

Abstract-

Page 2, line 32 – suggest revising the methods section to include a sentence describing the microbiome analysis. I also suggest revising the structure of the abstract to separate the purpose, methods and results sections.

We appreciate the reviewers’ suggestion and have added a sentence describing microbiome methods. We also modified the abstract. However, because of the multi-faceted nature of the project we feel that separating the sections in the abstracts would result in confusion and would prefer to have the methods and results of each part of the project follow each other.

Page 3, lines 54-56- The authors described the impact of TD on military operations/performance but did not provide any results related to the topic. Suggest providing some evidence (Ex- duty days lost).

“These findings illustrate the complex etiology of diarrhea amongst military personnel in deployed settings and its impacts on performance.”

We have added a statement to the first sentence of the abstract. Also we go into detail in the Introduction and the Results, so did not want to go into too much depth about it in the abstract, but we state that those data were collected.

Background-

Page 5, lines 91-93 (and page 4-5 paragraph)- It would be better if the authors can describe known evidence and research gap related to TD, then provide a specific hypothesis. In this paragraph, the author provides a good strong background on microbiota, function, and relationship to diseases/disorders in general. This paragraph needs at least some information relating to microbiota and TD.

“Thus, we hypothesize that gut microbial community composition can be a significant factor in symptomatic and sub-clinical syndromes.” Relating to TD or in general any illness?

We agree with the reviewers comments and have added to this section of the introduction to address their concerns.

Page 5, line 100- When authors mention of “illness”, does this mean any disease/disorder during deployment or just TD?

This refers to any illness during deployment, but we are focusing on those reporting TD.

Methods-

Page 6, line 110- Maybe it would be better to include a definition for “forward deployment” within brackets/parenthesis or use a more general term (Ex- deployed to a US military base on foreign soil/Central America).

We take the reviewer’s point. ‘Forward Operating Base (FOB)’ is the correct military description of this type of base. It is the equivalent of ‘frontline’ or ‘in harm’s way’ but we are not sure that these provide a better description. We have added ‘frontline’ to the description.

Page 6, lines 128-131- Did the questionnaire include a specific time duration? As an example, past events including signs and symptoms of diarrhea, respiratory, and febrile illnesses………during the past 30 days or 12 months? Also, I recommend including the questionnaire in the supplemental section (or provide the referenced Table SX).

The questionnaires described the time during their deployment and deployment durations varied, as well as symptoms and duration of diseases. We have added the pdf of the survey as a supplemental file.

Page 8, lines 160-161- I apologize for my unfamiliarity with the word “time in theater”, is this a specific room as a surgical theater? Please clarify?

We apologize, this is military nomenclature. To clarify, we have changed it to ‘deployment duration.’

Did the authors have the following baseline information? time since deployment, history of prior diarrheal infections, race,. risk factors contributing to the transmission of pathogenic bacteria within the deployment site (hygienic behavior assessment, availability of resources, etc.)

use of other medication (laxatives), use of probiotics (yogurt)?

These are all great points. Unfortunately we do not have those data. We mention that it is a well-appointed base in the discussion’s first sentence , but we did not detect significance with self-reported (iPod survey) behavior. As an example, surprisingly, Lactobacillus is frequently higher in subjects who are recording zero yogurt intake than those who are reporting yogurt consumption, and for those who recorded consumption, there was not a positive trend with increased consumption. The available metadata are incorporated into the supplemental visualization artifacts, and we’ve added text to better explain how to examine these data.

Page 8, lines 165-166- How did the authors treat multi-pathogen infections (co-infections) when calculating incidence for each pathogen?

Incidence of TD was calculated by a positive detection of any pathogen, so multi-pathogen detection was counted as a single event in the incidence. Numbers were too low to calculate individual pathogen incidences. The data show the number and percentages of tested stools that were multi-pathogen with 2-6 pathogens detected. The clinical incidence is likely much higher than our reported incidence, but we only received reports and samples of more severe cases and thus lower numbers. Mild TD is commonly under-reported. 

Page 9, lines 207-209- Did the authors use any kind of buffer to store the fecal samples (RNAlater)?

No, it was immediately frozen at -80 degrees C and was maintained at that temperature until sample processing.

Page 10, lines 221-227- Suggest adding a sub-heading

Sub-heading added.

Page 11, lines 239-241- Suggest including a Chao1 graph to the results section if possible

We calculated Chao1 alpha diversity for the microbiome data and added text (and modified figure S1) to reflect this. Unfortunately, as with the other alpha diversity tests, there were not significant results.

Results-

Page 15, Table 2- suggest specifying the duration of the assessment (Ex- during the past 12 months?) 

This was the end-of-deployment questionnaire described in the Methods. The length of deployment varied.

Page 17, Table 3- Earlier when describing other illness, it was not clear that the symptoms were associated with diarrhea or due to unrelated illnesses. Please clarify this in the methods section/other sections.

We collected data on any illness, as described in the methods, but we focused on illnesses associated with diarrhea in the Results and Table 3. We have added text to the results to clarify this.

Page 19-20, Table 4- Please provide a footnote with abbreviations and definitions

We have added definitions as a footnote to Tables 4 and 6.

Pages 23-24- Suggest including heat maps, diversity plots and figure representing relative abundance by phyla for groups (TD+ vs TD- summarized across all participants)

We did generate heatmaps of relative abundances during this process, however, they show an uninteresting result-most of the samples cluster by subject, including the specific time points of traveler's diarrhea. Collapsing the data into averages (i.e., first averaging a subject's time points to a subject average, and then averaging the groups that acquired TD versus those that did not) is possible, however, we think a lot of the data could be explored (at different taxonomic levels, or with different aspects of the metadata) via the QIIME2 artifact visualization files, which can be viewed directly in a browser without special software. We've added some details about how the reader can explore these data to the text to facilitate this.

Discussion-

Page 30, lines 588-591- Suggest revising the sentence as I don’t see the relevance of IBD and TD the way it is presented here.

We were trying to provide the reader with as much information as we could find regarding Ruminococcaceae UCG-013 and TD and human GI issues. We have altered the text to reflect this, rather than insinuating a causal relationship.

Page 31, lines 594-597- Is it possible that the changes in the gut microbiota due to doxycycline and other antimalarial prophylactics might contribute to the risk of TD? Please include past research evidence on this subject. It is possible participants taking such medication already have a disrupted gut microbiota.

As we discussed above, our sampling did not allow us to determine if doxycycline affected the subject’s microbiome or their susceptibility to diarrhea. We did not have the ability to have a control group that did not take doxycycline since all personnel are required to take doxycycline as a prophylactic antimalarial when deployed to locations where the risk of malaria is high. We have expanded our discussion to address the reviewer’s points.

Also, please briefly discuss the strengths of the study in the discussion section (controlled diet, similar physical activity levels, etc.).

 We thank the reviewer for the opportunity to highlight the strengths of this study. We are extremely proud of it and it is the result of many years of hard work and negotiation with Base Commanders. We have added a brief discussion of the study’s strengths to the Discussion.

Conclusion- Suggest revising the conclusion to include a summary of findings across all participants with TD compared to those without TD (or before and after TD results of individuals collectively).

We thank the reviewer for the suggestion. We have noted the commonalities that we detected quantitatively or observed in the longitudinal data.

6. PLOS authors have the option to publish the peer review history of their article (what does this mean?). If published, this will include your full peer review and any attached files.

Do you want your identity to be public for this peer review? For information about this choice, including consent withdrawal, please see our Privacy Policy.

Reviewer #1: No

Reviewer #2: No

Reviewer #3: No

Reviewer #4: Yes: Akemi T Wijayabahu

---

## [Decision Letter · Decision Letter 1]

1 Jul 2020

PONE-D-19-31202R1

Epidemiology and associated microbiota changes in deployed military personnel at high risk of traveler's diarrhea

PLOS ONE

Dear Dr. Goodson,

Thank you for submitting your manuscript to PLOS ONE. After careful consideration, we feel that it has merit but does not fully meet PLOS ONE’s publication criteria as it currently stands. Therefore, we invite you to submit a revised version of the manuscript that addresses the points raised during the review process.

All four reviewers and I are in agreement that the manuscript is much improved. Reviewers 2 and 4 have a few minor, helpful suggestions that should be incorporated into a revised document.

We look forward to receiving your revised manuscript.

Kind regards,

Brenda A Wilson, Ph.D.

Academic Editor

PLOS ONE

Reviewers' comments:

Reviewer's Responses to Questions

**Comments to the Author**

1. If the authors have adequately addressed your comments raised in a previous round of review and you feel that this manuscript is now acceptable for publication, you may indicate that here to bypass the “Comments to the Author” section, enter your conflict of interest statement in the “Confidential to Editor” section, and submit your "Accept" recommendation.

Reviewer #1: All comments have been addressed

Reviewer #2: (No Response)

Reviewer #3: All comments have been addressed

Reviewer #4: (No Response)

2. Is the manuscript technically sound, and do the data support the conclusions?

Reviewer #1: Yes

Reviewer #2: Yes

Reviewer #3: Yes

Reviewer #4: Yes

3. Has the statistical analysis been performed appropriately and rigorously? 

Reviewer #1: Yes

Reviewer #2: Yes

Reviewer #3: Yes

Reviewer #4: Yes

4. Have the authors made all data underlying the findings in their manuscript fully available?

Reviewer #1: Yes

Reviewer #2: Yes

Reviewer #3: (No Response)

Reviewer #4: Yes

5. Is the manuscript presented in an intelligible fashion and written in standard English?

Reviewer #1: Yes

Reviewer #2: Yes

Reviewer #3: Yes

Reviewer #4: Yes

6. Review Comments to the Author

Reviewer #1: The authors are congratulated for their execution and presentation of this important study. All of my previous comments have been satisfactorily addressed. Only 2 minor comments:

Abstract: Please add a statement describing the relevant results that support the statement “…and its impact on job performance”.

Discussion states “Our results are consistent with these prior observations, as we detect no differences for alpha diversity between TD+ and TD- subjects. It should be emphasized that while our results are statistically significant, the number of independent samples are small (N=4 TD- subjects, N=7 TD+ subjects).” Should the second sentence read “not statistically significant”?

Reviewer #2: (No Response)

Reviewer #3: Thank you for the authors for addressing all my comments. the manuscript now is accepted from my side with no further comments or modification.

Reviewer #4: Dear authors,

Thank you for being responsive to my recommendations and for the clarifications. I do not have any major concerns for the revised manuscript. It is much clearer and easier to understand.

Best wishes,

Akemi W.

Minor Comments:

Comment 1, Page 2– Suggest revising the following sentence to include explanations for each of the abbreviations that are mentioned for the first time.

Suggested revision: “We conducted a passive surveillance study of all cases of diarrhea reporting to the medical unit with 152 total cases and a similar pattern of etiology with 52/152 enteroaggregative E. coli (EAEC), 50/152 enteroinvasive E. coli (EIEC), and 35/152 enteropathogenic E. coli (EPEC) as the most prevalent pathogens detected, and ……”

Introduction-

Comment 2, Page 5- Evidence related to microbiome and TD is not sufficient, the research gap not identified, and the hypothesis is still not specific for the current research.

Of note, David et al, 2015 was the only study cited. Please consider including additional literature evidence on the relationship between microbiome and TD

Comment 3, Page 5- recommend clarifying what “illness” is in the manuscript.

Suggested revision: “After enrollment and collection of baseline samples, subjects were contacted weekly to assess the presence of any ailment specifically diarrheal diseases and if ill, ….”

Discussion-

Comment 4-Should it be just recall bias?

Suggested revision: “Similarly, questionnaires performed at the end of a person’s deployment may not accurately capture what they experienced because of recall bias.”

7. PLOS authors have the option to publish the peer review history of their article (what does this mean?). If published, this will include your full peer review and any attached files.

Reviewer #1: No

Reviewer #2: No

Reviewer #3: No

Reviewer #4: **Yes: **Akemi T Wijayabahu

---

## [Author Response · Author response to Decision Letter 1]

10 Jul 2020

Reviewer #1: The authors are congratulated for their execution and presentation of this important study. All of my previous comments have been satisfactorily addressed. Only 2 minor comments:

We thank the reviewer for their kind words and for their comments.

Abstract: Please add a statement describing the relevant results that support the statement “…and its impact on job performance”.

We have added results to support this statement, and modified the abstract to keep it within the 300 word limit.

Discussion states “Our results are consistent with these prior observations, as we detect no differences for alpha diversity between TD+ and TD- subjects. It should be emphasized that while our results are statistically significant, the number of independent samples are small (N=4 TD- subjects, N=7 TD+ subjects).” Should the second sentence read “not statistically significant”?

We thank the reviewer for this change. We agree that for the results described in this paragraph should be changed to ‘not statistically significant’. We also discuss our small sample size when we comment on statistically significant results later in the Discussion.

Reviewer #2: (No Response)

Reviewer #3: Thank you for the authors for addressing all my comments. the manuscript now is accepted from my side with no further comments or modification.

We thank the reviewer for their time and effort.

Reviewer #4: Dear authors,

Thank you for being responsive to my recommendations and for the clarifications. I do not have any major concerns for the revised manuscript. It is much clearer and easier to understand.

Best wishes,

Akemi W.

We thank you for your comments and suggestions.

Minor Comments:

Comment 1, Page 2– Suggest revising the following sentence to include explanations for each of the abbreviations that are mentioned for the first time.

Suggested revision: “We conducted a passive surveillance study of all cases of diarrhea reporting to the medical unit with 152 total cases and a similar pattern of etiology with 52/152 enteroaggregative E. coli (EAEC), 50/152 enteroinvasive E. coli (EIEC), and 35/152 enteropathogenic E. coli (EPEC) as the most prevalent pathogens detected, and ……”

We have added explanations for each of the abbreviations as suggested by the reviewer, and modified the abstract to keep it within the 300 word limit.

Introduction-

Comment 2, Page 5- Evidence related to microbiome and TD is not sufficient, the research gap not identified, and the hypothesis is still not specific for the current research.

Of note, David et al, 2015 was the only study cited. Please consider including additional literature evidence on the relationship between microbiome and TD

We thank the reviewer for these inputs.

We have added additional literature references supporting a relationship between the gut microbiome and TD. We delve into these further in the Discussion where we compare these papers to our results, and so did not feel that we should repeat an in-depth summary of the literature here, other than providing a broad summary of their results.

We have made the hypothesis specific for the current research by adding ‘can contribute to susceptibility or resilience to TD’ to the final sentence of the paragraph.

We have further identified the research gap at the beginning of the next paragraph by restating why a better understanding of how microbial community composition affects military personnel experiencing diarrhea during deployment is important.

Comment 3, Page 5- recommend clarifying what “illness” is in the manuscript.

Suggested revision: “After enrollment and collection of baseline samples, subjects were contacted weekly to assess the presence of any ailment specifically diarrheal diseases and if ill, ….”

We have added clarification based on the reviewers suggestion.

Discussion-

Comment 4-Should it be just recall bias?

Suggested revision: “Similarly, questionnaires performed at the end of a person’s deployment may not accurately capture what they experienced because of recall bias.”

Yes, we agree. We changed the sentence to the reviewer’s suggestion.

---

## [Editor Report · Decision Letter 2]

14 Jul 2020

Epidemiology and associated microbiota changes in deployed military personnel at high risk of traveler's diarrhea

PONE-D-19-31202R2

Dear Dr. Goodson,

We’re pleased to inform you that your manuscript has been judged scientifically suitable for publication and will be formally accepted for publication once it meets all outstanding technical requirements.

Kind regards,

Brenda A Wilson, Ph.D.

Academic Editor

PLOS ONE

Additional Editor Comments (optional):

None
---

## [Editor Report · Acceptance letter]

17 Jul 2020

PONE-D-19-31202R2 

Epidemiology and associated microbiota changes in deployed military personnel at high risk of traveler's diarrhea 

Dear Dr. Goodson:

I'm pleased to inform you that your manuscript has been deemed suitable for publication in PLOS ONE. Congratulations! Your manuscript is now with our production department. 

Kind regards, 

on behalf of

Dr. Brenda A Wilson 

Academic Editor

PLOS ONE